# Error Analysis of Discrete Flow with Generator Matching

**Zhengyan Wan**[1]  **Yidong Ouyang**[2]  **Qiang Yao**[1]  **Liyan Xie**[3]  **Fang Fang**[†1]  **Hongyuan Zha**[†4]  **Guang Cheng**[2]

## Abstract

Discrete flow models offer a powerful framework for learning distributions over discrete state spaces and have demonstrated superior performance compared to the discrete diffusion models. However, their convergence properties and error analysis remain largely unexplored. In this work, we develop a unified framework grounded in stochastic calculus theory to systematically investigate the theoretical properties of discrete flow models. Specifically, by leveraging a Girsanov-type theorem for the path measures of two continuous-time Markov chains (CTMCs), we present a comprehensive error analysis that accounts for both transition rate estimation error and early stopping error. In fact, the estimation error of transition rates has received little attention in existing works. Unlike discrete diffusion models, discrete flow incurs no initialization error caused by truncating the time horizon in the noising process. Building on generator matching and uniformization, we establish non-asymptotic error bounds for distribution estimation without the boundedness condition on oracle transition rates. Furthermore, we derive a faster rate of total variation convergence for the estimated distribution with the boundedness condition, yielding a nearly optimal rate in terms of sample size. Our results provide the first error analysis for discrete flow models. We also investigate model performance under different settings based on simulation results.

---

[1]School of Statistics, East China Normal University [2]Department of Statistics and Data Science, University of California, Los Angeles [3]Department of Industrial and Systems Engineering, University of Minnesota [4]School of Data Science, The Chinese University of Hong Kong, Shenzhen. Correspondence to: Fang Fang <ffang@sfs.ecnu.edu.cn>, Hongyuan Zha <zhahy@cuhk.edu.cn>.

*Proceedings of the 43rd International Conference on Machine Learning*, Seoul, South Korea. PMLR 306, 2026. Copyright 2026 by the author(s).

## 1. Introduction

Discrete diffusion models have achieved significant progress in large language models (Nie et al., 2025; Zhu et al., 2025; Zhao et al., 2025; Yang et al., 2025). By learning the time reversal of the noising process of a continuous-time Markov chain (CTMC), the models transform a simple distribution (e.g., uniform (Hoogeboom et al., 2021; Lou et al., 2024) and masked (Ou et al., 2025; Shi et al., 2024; Sahoo et al., 2024)) that is easy to sample to the data distribution that has discrete structures. Discrete flow models (Campbell et al., 2024; Gat et al., 2024; Shaul et al., 2025) provide a flexible framework for learning generating transition rate analogous to continuous flow matching (Albergo & Vanden-Eijnden, 2023; Liu et al., 2023; Lipman et al., 2023).

Recent theoretical analysis for discrete diffusion models has emerged (Chen & Ying, 2024; Zhang et al., 2025; Liang et al., 2025a;b; Pham et al., 2025; Ren et al., 2025a;b). To obtain the transition rate in the reversed process, the concrete scores in these analyses are obtained by minimizing the concrete score entropy introduced in (Lou et al., 2024; Benton et al., 2024b). In those works, the distribution errors of discrete diffusion models are divided into three parts: (a) initialization error from truncating the time horizon in the noising process; (b) concrete score estimation error; (c) discretization error from sampling algorithms. In our paper, we aim to investigate the theoretical properties of the discrete flow-based models using the generator matching training objective (Holderrieth et al., 2025) and the uniformization sampling algorithm (Chen & Ying, 2024), which offers zero initialization error and discretization error. Our analysis takes transition rate estimation error into account instead of imposing a stringent condition on it in previous works, which is related to the early stopping parameter. We decompose the estimation error into stochastic error and approximation error, and then balance the stochastic error and early stopping error by choosing the early stopping parameter. Our stochastic error bound is aligned with the results in continuous flow (Gao et al., 2024b); the early stopping error bound matches the recent result in discrete diffusion (Zhang et al., 2025). Furthermore, we present a comprehensive error analysis for the neural network class with the ReLU activation function, by controlling stochastic error, approximation error and early stopping error simultaneously. To the best of our knowledge, this is the first theoretical error analysis

for discrete flow models. We additionally conduct simulation experiments to investigate empirical performance under various settings.

The main contributions in this paper are summarized as follows.

1. Parallel to the continuous flow matching framework, we establish a unified theoretical framework for discrete flow-based models via generator matching, which enables us to control the KL divergence between the path measures of two CTMCs naturally. In the sampling stage, we use the uniformization algorithm to sample in an exact way.

2. We establish the non-asymptotic error bound for distribution error in discrete flow models, by taking estimation error into consideration, which is less explored in existing works. There are three sources of error in our framework: (a) stochastic error from estimation through empirical risk minimization; (b) approximation error of the selected function class; (c) early stopping error. We provide the total variation error bound for the estimated distribution without imposing conditions on the data distribution and the oracle transition rate. Additionally, under the boundedness condition of the oracle transition rate, we derive a faster convergence rate of the total variation, which is nearly optimal in terms of the sample size.

Due to space limitations, some additional results and all technical proofs are deferred to the Appendix.

NOTATION. Let $[N] = \{1, 2, \ldots, N\}$ for a positive integer $N$. We use $\dot{\kappa}_t$ to denote the time derivative of a function $\kappa_t$ of $t$. For a $\mathcal{D}$-dimensional vector $z$, let $z^d$ and $z^{\backslash d}$ denote the $d$-th element of the vector $z$ and the $(\mathcal{D}-1)$-dimensional vector $(z^1, \ldots, z^{d-1}, z^{d+1}, \ldots, z^{\mathcal{D}})^\top$. We use $\mathbb{1}(\cdot)$ to denote an indicator function. We denote the Hamming distance of two vectors $z, x$ by $\mathrm{d}^H(z, x)$. For two quantities $x, z$, define the Kronecker delta $\delta_x(z)$ satisfying $\delta_x(z) = 1$ if $x = z$ and $\delta_x(z) = 0$ if $x \neq z$. We say $a = \mathcal{O}(b)$ if $a \leq cb$ for a positive constant $c$. In this paper, some universal constants $C, c > 0$ are allowed to vary from line to line.

## 2. Theoretical Background on Discrete Flow-Based Models

In this section, we introduce some theoretical background about continuous-time Markov chains (Norris, 1998) and discrete flow-based models (Campbell et al., 2024; Gat et al., 2024; Shaul et al., 2025; Wang et al., 2025).

### 2.1. Continuous-Time Markov Chain as Stochastic Integral

In this subsection, we consider the probability space $(\Omega, \mathcal{F}, \mathbb{P})$. We formally define continuous-time Markov chains (CTMCs) as follows.

**Definition 1** (CTMC). *Consider a $\mathcal{D}$-dimensional finite state space $\mathcal{S}^{\mathcal{D}}$, where $\mathcal{S} = \{1, 2, \ldots, |\mathcal{S}|\}$. Let $u_t(z, x)_{z, x \in \mathcal{S}^{\mathcal{D}}}$ be a (time-dependent) rate matrix that satisfying: (a) $u_t(z, x)$ is continuous in $t$; (b) $u_t(z, x) \geq 0$ for any $z \neq x$; (c) $\sum_{z \in \mathcal{S}^{\mathcal{D}}} u_t(z, x) = 0$ for any $x \in \mathcal{S}^{\mathcal{D}}$. We call a process $\{X(t)\}_{t \geq 0}$ a continuous time Markov chain with the transition rate matrix $u_t(z, x)_{z, x \in \mathcal{S}^{\mathcal{D}}}$ and the natural filtration $\mathcal{F}_t = \sigma(\{X(s) : 0 \leq s \leq t\})$ if it satisfies that for any $z, x \in \mathcal{S}^{\mathcal{D}}$*

1. *the transition rate is the generator of CTMC: $\mathbb{P}(X(t + h) = z|X(t) = x) = \delta_x(z) + u_t(z, x)h + o(h)$;*

2. *it has Markov property: $\mathbb{P}(X(t + h) = z|\mathcal{F}_t) = \mathbb{P}(X(t + h) = z|X(t))$.*

Given a rate matrix $u_t$ with uniformly bounded entries, the following uniformization technique allows us to construct a CTMC with $u_t$ through a Poisson process, which offers us a sampling algorithm without discretization error (see, e.g., Ren et al., 2025a; Chen & Ying, 2024).

**Proposition 1** (Uniformization). *Assume that $u_t(z, x)_{z, x \in \mathcal{S}^{\mathcal{D}}}$ is a rate matrix satisfying that (a) $-u_t(x, x) \leq M$ for any $x \in \mathcal{S}^{\mathcal{D}}$; (b) $u_t(z, x)$ is $L$-Lipschitz continuous in $t$ for any pair $(z, x) \in \mathcal{S}^{\mathcal{D}} \times \mathcal{S}^{\mathcal{D}}$. Suppose that $T_1, T_2, \ldots$ are the arrival times of a Poisson process $N(t)$ with rate $M$. Let $X(t)$ be a jump process with initial distribution $p_0$ and natural filtration $\mathcal{F}_t$ such that at $t = T_1, T_2, \ldots$, the process $X(t)$ jumps to position $z \neq X(t-)$ with probability $u_t(z, X(t-))/M$. Then $X(t)$ is a Markov process satisfying for $z \neq x$,*

$$\mathbb{P}(X(t + h) = z|X(t) = x) = u_t(z, x)h + R_t,$$

*where $R_t \leq (M^2 + L)h^2 = O(h^2)$. Thus, $X(t)$ is a CTMC with the rate $u_t$.*

Here, the remainder $R_t$ is uniformly bounded for any $t$ and $z \neq x$ under the assumptions in Proposition 1, which is crucial for developing the theory of CTMC.

To rewrite a CTMC $X(t)$ as a stochastic integral, we define a random measure $N((t_1, t_2], A)$ associated with the CTMC $X(t)$.

**Definition 2** (Random Measure Associated with CTMC). *Suppose that $X(t)$ is a CTMC with rate $u_t$ and natural filtration $\mathcal{F}_t$. Define the random measure associated with $X(t)$ and $A \subseteq \mathcal{S}^{\mathcal{D}}$ as*

$$N((t_1, t_2], A) = \#\{t_1 < s \leq t_2; \Delta X(s) \neq 0, X(s) \in A\},$$

*where $\Delta X(t) = X(t) - X(t-)$.*

Suppose that $N((t_1, t_2], A)$ is the random measure associated with a CTMC $X(t)$, where $X(t)$ is constructed by uniformization with the Poisson process $N(t)$. Then $N(t, A) \triangleq N((0, t], A) \leq N(t) < \infty$ a.s. for any $t \geq 0$ and $A \subseteq \mathcal{S}^{\mathcal{D}}$ (see Lemma 2.3.4 in Applebaum, 2009). Thus, similar to the Poisson random measure associated with a Lévy process (see Section 2.3.2 in Applebaum, 2009), there are some equivalent representations:

$$N((t_1, t_2], A) = \sum_{n \in \mathbb{N}} \delta_{(t_1, t_2]}(T_n^A),$$

where $\{T_n^A\}_{n \in \mathbb{N}}$ are arrival times of the counting process $N(t, A)$. Consequently, $X(t)$ can be written as the following stochastic integral:

$$\begin{aligned}
X(t) &= X(0) + \sum_{n \in \mathbb{N}} \Big[ X(T_n) - X(T_{n-1}) \Big] \delta_{[0, t]}(T_n) \\
&= X(0) + \int_0^t \int_{\mathcal{S}^{\mathcal{D}}} (z - X(s-)) N(ds, dz),
\end{aligned} \tag{1}$$

where $\{T_n\}_{n \in \mathbb{N}}$ are arrival times of $N(t, \mathcal{S}^{\mathcal{D}})$. Here, we formally define the integrator $N(t, A)$ in the CTMC representation as Definition 2, which is clearer and more formal than the related definition compared to Proposition 3.2 in (Ren et al., 2025a).

**Remark 1** (Comparison to Lévy-Itô Decomposition). *Recall that a Lévy process $X(t)$ has the following Lévy-Itô decomposition (see Theorem 2.4.16 in Applebaum, 2009):*

$$\begin{aligned}
X(t) &= X(0) + bt + B(t) \\
&\quad + \int_{|x|<1} x \tilde{N}(t, dx) + \int_{|x| \geq 1} x N(t, dx),
\end{aligned}$$

*where $B(t)$ is a Brownian motion, $N(t, A)$ is the Poisson random measure associated with the Lévy process $X(t)$, and $\tilde{N}(t, A)$ is the compensator of $N(t, A)$. Here, the second argument in $N(t, A)$ is the jump size of the associated Lévy process at time $t$. However, the second argument of the random measure associated with a CTMC in Definition 2 is the position after jump at time $t$. Both the integrands in equation 1 and that in Lévy-Itô decomposition are jump size. Moreover, since CTMCs do not have independent increments in general, the random measure in Definition 2 might not independently scattered; that is, $N(t, A_1)$ and $N(t, A_2)$ are not necessarily independent for disjoint $A_1, A_2 \subseteq \mathcal{S}^{\mathcal{D}}$, which is not a Poisson random measure. There is no need to use the independently scattered property in our technical proofs.*

Given a CTMC $X(t)$ with transition rate $u_t$ and marginal densities $\{p_t\}_{t \geq 0}$ w.r.t. the counting measure, the following proposition demonstrates that the marginal densities $p_t$

satisfy the following Kolmogorov forward equation (a.k.a. continuity equation).

**Proposition 2** (Kolmogorov Forward Equation). *The CTMC $X(t)$ satisfies the following equation:*

$$\dot{p}_t(x) = \underbrace{\sum_{z \neq x} u_t(x, z) p_t(z)}_{\text{Incoming Flux}} - \underbrace{\sum_{z \neq x} u_t(z, x) p_t(x)}_{\text{Outgoing Flux}}.$$

In our work, we say $u_t$ can generate the probability path $p_t$, if it satisfies the above Kolmogorov equation.

For better presentation, some additional results for CTMCs, including the compensator of the random measure and a Girsanov-type theorem, are deferred to Section C.

## 2.2. Discrete Flow-Based Models

We aim to learn a transition rate $u_t$ of a CTMC $X(t)$ that can transport from a source distribution $p_0$ to a target data distribution $p_1$. To obtain such a transition rate for sampling, a natural method is to learn the conditional expectation of the conditional transition rate $u_t(z, x|x_1)$ that generates the conditional probability path $p_{t|1}(\cdot|x_1)$, since $u_t(z, x) = \mathbb{E}[u_\mathbf{t}(z, x|X(1))|X(\mathbf{t}) = x, \mathbf{t} = t]$ can generate the target probability path $p_t$ (see Proposition 3.1 of Campbell et al., 2024), i.e., it satisfies the Kolmogorov forward equation, where $X(\mathbf{t}) \sim p_{\mathbf{t}|1}(\cdot|X(1))$ given $X(1)$ and $\mathbf{t} \sim \mathcal{U}([0, 1])$. Therefore, we are free to define the conditional probability path and the conditional transition rate.

It is worth noting that in the sampling stage, considering a $|\mathcal{S}|^{\mathcal{D}}$-dimensional vector-valued function $(u_t(z, x))_{z \in \mathcal{S}^{\mathcal{D}}}$ of current time $t$ and state $x$ is intractable when $\mathcal{D}$ is relatively large. To handle such a high-dimensional scenario, a common approach is to construct a coordinate-wise conditional probability path and transition rate, that is,

$$\begin{aligned}
p_{t|1}(x|x_1) &= \prod_{d=1}^{\mathcal{D}} p_{t|1}^d(x^d|x_1^d), \\
\text{and } u_t(z, x|x_1) &= \sum_{d=1}^{\mathcal{D}} \delta_{x^{\backslash d}}(z^{\backslash d}) u_t^d(z^d, x^d|x_1^d),
\end{aligned} \tag{2}$$

which means that the elements of the vector $X(t)$ are independent conditional on $X(1)$. Here, $u_t^d(z^d, x^d|x_1^d)$ is the conditional transition rate that generates the conditional probability path $p_{t|1}^d$. A popular choice of probability path and the associated conditional transition rate used in the previous works (Campbell et al., 2024; Gat et al., 2024) is

$$\begin{aligned}
p_{t|1}^d(x^d|x_1^d) &= (1 - \kappa_t) p_0^d(x^d) + \kappa_t \delta_{x_1^d}(x^d); \\
u_t^d(z^d, x^d|x_1^d) &= \frac{\dot{\kappa}_t}{1 - \kappa_t} (\delta_{x_1^d}(z^d) - \delta_{x^d}(z^d)),
\end{aligned} \tag{3}$$

where $\kappa_t : [0, 1] \to [0, 1]$ is a non-decreasing function satisfying $\kappa_0 = 0$ and $\kappa_1 = 1$. Note that the conditional

transition rate will blow up as $t \to 1^-$. Thus, in the sampling stage, we employ the early stopping technique; that is, we only consider the time interval $[0, 1 - \tau]$ for a sufficiently small positive parameter $\tau$. We will discuss this time singularity in Section D.1.

After defining the conditional path and rate, the unconditional transition rate is given by

$$
\begin{aligned}
u_t(z, x) &= \sum_{d=1}^{\mathcal{D}} \delta_{x \backslash d}(z^{\backslash d}) \sum_{x_1^d} u_t^d(z^d, x^d | x_1^d) p_{1|t}^d(x_1^d | x) \\
&\triangleq \sum_{d=1}^{\mathcal{D}} \delta_{x \backslash d}(z^{\backslash d}) u_t^d(z^d, x),
\end{aligned}
$$
(4)

where $p_{1|t}^d(x_1^d | x) = \sum_{x_1^{\backslash d}} p_{1|t}(x_1 | x)$. Therefore, it suffices to consider a sparse rate matrix $u_t$ satisfying $u_t(z, x) = 0$ for any $z, x \in \mathcal{S}^{\mathcal{D}}$ with Hamming distance $\mathrm{d}^H(z, x) > 1$.

In our work, we consider both masked and uniform source distributions, which are commonly used in practice (Campbell et al., 2024; Wang et al., 2025; Deng et al., 2025).

**Training via Bregman divergence.** Let $\tau \in (0, 1/2)$ be an early stopping parameter. Suppose that we have i.i.d. samples $\{\mathbf{t}_i, X_i(1)\}_{i \in [n]}$, and $X_i(\mathbf{t}_i) \sim p_{\mathbf{t}_i | 1}(\cdot | X_i(1))$ for each $i \in [n]$, where $\mathbf{t}_i$ samples from the uniform distribution $\mathcal{U}([0, 1 - \tau])$. Denote $\mathbb{D}_n = \{Z_i\}_{i \in [n]} = \{(\mathbf{t}_i, X_i(\mathbf{t}_i), X_i(1))\}_{i \in [n]}$ and $v(x, Z_i) = u_{\mathbf{t}_i}(x, X_i(\mathbf{t}_i) | X_i(1))$, where $u_t(z, x | x_1)$ is a conditional transition rate that generates the conditional probability path $p_{t|1}(x | x_1)$. Denote the Bregman divergence with a convex function $F(\cdot)$ as $D_F$, which is defined as

$$
D_F(a || b) = F(a) - F(b) - \langle \nabla F(b), a - b \rangle.
$$

Define the following quantity:

$$
\begin{aligned}
&\mathcal{I}(u, z, \mathbf{t}, X(\mathbf{t}), X(1)) \\
&\triangleq -u_{\mathbf{t}}(z, X(\mathbf{t}) | X(1)) \log u_{\mathbf{t}}(z, X(\mathbf{t})) + u_{\mathbf{t}}(z, X(\mathbf{t})).
\end{aligned}
$$

Inspired by generator matching (Holderrieth et al., 2025), we consider the following rate estimator through empirical risk minimization (ERM) with a function class $\mathcal{G}_n$:

$$
\begin{aligned}
\hat{u} &= \underset{u \in \mathcal{G}_n}{\arg\min} \frac{1}{n} \sum_{i=1}^{n} \sum_{z \neq X_i(\mathbf{t}_i)} D_F(v(z, Z_i) || u_{\mathbf{t}_i}(z, X_i(\mathbf{t}_i))) \\
&= \underset{u \in \mathcal{G}_n}{\arg\min} \frac{1}{n} \sum_{i=1}^{n} \sum_{z \neq X_i(\mathbf{t}_i)} \mathcal{I}(u, z, \mathbf{t}_i, X_i(\mathbf{t}_i), X_i(1)) \\
&\triangleq \underset{u \in \mathcal{G}_n}{\arg\min} \mathcal{L}_n(u),
\end{aligned}
$$
(5)

where we take $F(x) = x \log x$ and the summation over $\{z : z \neq X_i(\mathbf{t}_i)\}$ has only $O(\mathcal{D}|S|)$ complexity due to the sparsity of the rate matrix. Here, the training objective is finite if we choose an appropriate function class (e.g. equation 7 in Section 3).

**Sampling via uniformization.** For accurate sampling without discretization error, following the uniformization argument introduced in Proposition 1, we consider the uniformization Algorithm 1 in the Appendix (also see, e.g. Algorithm 1 in Chen & Ying, 2024). Here, the discretization used in Algorithm 1 (see Section B in the Appendix) can reduce the average sampling steps compared to using a uniform bound $M \geq \sup_{t \in [0, 1-\tau]} \sum_{z \neq x} \hat{u}_t(z, x)$ for a large time interval $[0, 1 - \tau]$.

## 3. Main Results

In this section, we establish the error bound for discrete flow-based models. We will first introduce some additional notation. Then, we will present a result for bounding the KL divergence of the marginal distributions of two CTMCs. After that, we will derive the error bounds for stochastic error, approximation error and early stopping error without boundedness condition on the oracle rate. Finally, we establish a fast convergence rate for the distribution error under the boundedness condition.

### 3.1. Additional Notations

We first introduce some additional notation. Similar to the training objective (Equation 5), we denote the oracle transition rate as

$$
\begin{aligned}
u^0 &= \underset{u}{\arg\min} \mathbb{E}\left\{ \sum_{z \neq X(\mathbf{t})} \mathcal{I}(u, z, \mathbf{t}, X(\mathbf{t}), X(1)) \right\} \\
&\triangleq \underset{u}{\arg\min} \mathcal{L}(u).
\end{aligned}
$$

where $Z = (\mathbf{t}, X(\mathbf{t}), X(1))$ is a test point independent of the data $\mathbb{D}_n$. Through marginalization trick (Lipman et al., 2024), we have $u_t^0(z, x) = \mathbb{E}[u_{\mathbf{t}}(z, x | X(1)) | X(\mathbf{t}) = x, \mathbf{t} = t]$. We also define the best approximation in the function class $\mathcal{G}_n$ as

$$
\begin{aligned}
u^* &= \underset{u \in \mathcal{G}_n}{\arg\min} \mathbb{E}\left\{ \sum_{z \neq X(\mathbf{t})} \mathcal{I}(u, z, \mathbf{t}, X(\mathbf{t}), X(1)) \right\} \\
&\triangleq \underset{u \in \mathcal{G}_n}{\arg\min} \mathcal{L}(u).
\end{aligned}
$$

For notational simplicity, we define

$$
\begin{aligned}
g(u, z, Z) &\triangleq -u_{\mathbf{t}}(z, X(\mathbf{t}) | X(1)) \log \frac{u_{\mathbf{t}}(z, X(\mathbf{t}))}{u_{\mathbf{t}}^0(z, X(\mathbf{t}))} \\
&\quad + u_{\mathbf{t}}(z, X(\mathbf{t})) - u_{\mathbf{t}}^0(z, X(\mathbf{t})).
\end{aligned}
$$

Then, the approximation error of $u^*$ is

$$\mathcal{L}(u^*) - \mathcal{L}(u^0) = \mathbb{E}\Big[ \sum_{z \neq X(\mathbf{t})} g(u^*, z, Z) \Big]$$

$$= \inf_{u \in \mathcal{G}_n} \mathbb{E}\Big[ \sum_{z \neq X(\mathbf{t})} D_F\Big(u_{\mathbf{t}}^0(z, X(\mathbf{t})) \| u_{\mathbf{t}}(z, X(\mathbf{t}))\Big) \Big],$$

(6)

where the third equation we use the marginalization trick.

Since the oracle transition rate can be written as $u_t^0(z, x) = \sum_{d=1}^{\mathcal{D}} \delta_{x^{\backslash d}}(z^{\backslash d}) u_t^{0,d}(z^d, x)$, we consider the following matrix-valued function class:

$$\mathcal{G}_n = \Big\{ \{u_{\cdot}^d(s, \cdot)\}_{d \in \mathcal{D}, s \in \mathcal{S}} : \mathcal{S}^{\mathcal{D}} \times [0, 1] \to \mathbb{R}^{\mathcal{D} \times |\mathcal{S}|} :$$
$$|u_t^d(z^d, x)| \geq m_n, u_{\cdot}^d(s, \cdot) \in \mathcal{G}_n' \Big\},$$

(7)

where $\mathcal{G}_n'$ is a class of scalar-valued functions. Let $M_\tau = \sup_{t \in [0, 1-\tau]} \frac{\dot{\kappa}_t}{1-\kappa_t}$ be the upper bound of the conditional transition rate $u_t(z, x | x_1)$ for $\mathrm{d}^H(z, x) = 1$.

## 3.2. Bound for KL Divergence

We first present a general result for bounding the KL divergence between the marginal distributions of two CTMCs. Suppose that $X(t)$ has marginal distribution $p_t^X$ on the probability space $(\Omega, \mathcal{F}, \mathbb{P}^X)$ and marginal distribution $p_t^Y$ on the probability space $(\Omega, \mathcal{F}, \mathbb{P}^Y)$. According to Girsanov's theorem (Theorem 8), we can derive the bound for KL divergence of $p_t^X$ and $p_t^Y$.

**Theorem 1** (Bound for KL Divergence). *Suppose that $X(t)$ is a CTMC with natural filtration $\mathcal{F}_t$ and rate $u_t^X$ under measure $\mathbb{P}^X$; with rate $u_t^Y$ under measure $\mathbb{P}^Y$. Under the conditions in Theorem 8, the KL divergence between $p_t^X$ and $p_t^Y$ is*

$$D_{KL}(p_t^X \| p_t^Y)$$
$$\leq \mathbb{E}^X \Big\{ \int_0^t \sum_{x \neq X(s)} D_F(u_s^X(x, X(s)) \| u_s^Y(x, X(s))) ds \Big\}.$$

*where $D_F$ is the Bregman divergence with function $F(x) = x \log x$.*

Theorem 1 is *crucial* for developing the estimation error. Due to the sparsity of the transition rate matrix, intuitively, if $D_F(u_t^X(x, z) \| u_t^Y(x, z)) \leq \epsilon$ uniformly for $t \in [0, 1-\tau]$ and $(x, z)$ with $\mathrm{d}^H(x, z) = 1$, then $D_{KL}(p_{1-\tau}^X \| p_{1-\tau}^Y) \leq \mathcal{D}|\mathcal{S}|\epsilon$, which is linear in $\mathcal{D}$. In the next subsection, we will systematically perform an error analysis for our distribution estimation based on discrete flow models.

## 3.3. Error Analysis without Boundedness Condition

In this subsection, we establish the non-asymptotic error bound for the estimated distribution without imposing conditions on the data distribution and oracle transition rate.

Suppose that $p_t$ (resp. $\hat{p}_t$) is the marginal distribution of a CTMC with rate $u_t^0$ (resp. $\hat{u}_t$) at time $t$, where $p_0 = \hat{p}_0$. For random samples $\mathbb{D}_n$, according to Theorem 1, we have

$$\mathbb{E}_{\mathbb{D}_n}[D_{KL}(p_{1-\tau} \| \hat{p}_{1-\tau})] \leq (1-\tau)\mathbb{E}\Big\{ \sum_{z \neq X(\mathbf{t})} g(\hat{u}, z, Z) \Big\}$$
$$= \mathbb{E}_{\mathbb{D}_n}[\mathcal{L}(\hat{u}) - \mathcal{L}(u^0)].$$

**Estimation Error Decomposition.** The following Proposition gives us a standard error decomposition for empirical risk minimization, which decomposes the estimation error into stochastic error and approximation error.

**Proposition 3** (Estimation Error Decomposition 1). *For random sample $\mathbb{D}_n$, the excess risk of the estimator $\hat{u}$ through ERM (Equation 5) satisfies*

$$\mathbb{E}[\mathcal{L}(\hat{u}) - \mathcal{L}(u^0)] \leq \underbrace{\mathbb{E} \sup_{u \in \mathcal{G}_n} \Big| \mathcal{L}_n(u) - \mathcal{L}(u) \Big|}_{Stochastic\ Error}$$

$$+ \underbrace{\inf_{u \in \mathcal{G}_n} \mathbb{E}\Big[ \sum_{z \neq X(\mathbf{t})} D_F(u_{\mathbf{t}}^0(z, X(\mathbf{t}) \| u_{\mathbf{t}}(z, X(\mathbf{t}))) \Big]}_{Approximation\ Error}.$$

**Stochastic Error Bound.** Now, we establish the upper bound of the stochastic error in Proposition 3 using the empirical process theory. Before presenting our stochastic error bound, we first introduce the definition of uniform covering number.

**Definition 3** (Uniform Covering Number, (Jiao et al., 2023)). *Let $S$ be a subset of $\mathbb{R}^n$. Given a positive real number $\epsilon$, a subset $\mathcal{C}$ of $S$ is called an $\epsilon$-covering of $S$ w.r.t. the infinity norm if for any $x \in S$, there is $z \in \mathcal{C}$ such that $\|z - x\|_{L^\infty} < \epsilon$. The minimal cardinality $\mathcal{N}_n(\epsilon, S, L^\infty)$ of all possible $\mathcal{C}$ is called the covering number of $S$. For a given sequence $\mathbf{s} = \{x_i\}_{i=1}^n \in (\mathcal{X})^n$, let $\mathcal{F}|_{\mathbf{s}} = \Big\{ \Big( f(x_1), f(x_2), \ldots, f(x_n) \Big) : f \in \mathcal{F} \Big\}$. We define the covering number of $\mathcal{F}$ constrained on $\mathbf{s}$ as $\mathcal{N}_n(\epsilon, \mathcal{F}|_{\mathbf{s}}, L^\infty)$. The uniform covering number is defined as*

$$\mathcal{N}_n(\epsilon, \mathcal{F}, L^\infty) = \max \{\mathcal{N}_n(\epsilon, \mathcal{F}|_{\mathbf{s}}, L^\infty) : \mathbf{s} \in (\mathcal{X})^n\}.$$

**Theorem 2** (Stochastic Error). *The stochastic error in Proposition 3 has the following upper bound:*

$$\mathbb{E} \sup_{u \in \mathcal{G}_n} \Big| \mathcal{L}_n(u) - \mathcal{L}(u) \Big|$$

$$\leq C M_\tau (\log m_n^{-1} + 1) \mathcal{D}|\mathcal{S}| \sqrt{\frac{\log [\mathcal{D}|\mathcal{S}|\mathcal{N}_n(\frac{cm_n}{n}, \mathcal{G}_n', L^\infty)]}{n}}.$$

**Approximation Error Bound.** We consider using neural network (NN) functions with ReLU activation function to approximate the oracle transition rate $u^0$. To impose some

regularity condition on the oracle rate, we first introduce the Hölder class. For a finite constant $B_0 > 0$ and input dimension $d \in \mathbb{N}^+$, the Hölder class $\mathcal{H}^\beta([0,1]^d, B_0)$ is defined as

$$\mathcal{H}^\beta([0,1]^d, B_0) = \Big\{ f : [0,1]^d \to \mathbb{R}, \max_{\|\alpha\|_1 \leq s} \|\partial^\alpha f\|_\infty \leq B_0,$$

$$\max_{\|\alpha\|_1 = s} \sup_{x \neq y} \frac{|\partial^\alpha f(x) - \partial^\alpha f(y)|}{\|x - y\|_2^r} \leq B_0 \Big\},$$

where $\beta = r + s, s = \lfloor \beta \rfloor \in \mathbb{N}$, $\partial^\alpha = \partial^{\alpha_1} \cdots \partial^{\alpha_d}$ with $\alpha = (\alpha_1, \ldots, \alpha_d)^\top \in \mathbb{N}^d$ and $\|\alpha\|_1 = \sum_{i=1}^d \alpha_i$.

**Assumption 1** (Hölder Smoothness). *For each $d \in [\mathcal{D}]$ and $s \in \mathcal{S}$, there exists a function $\bar{u}^{0,d}(s, \cdot)$ belongs to the Hölder class $\mathcal{H}^\beta([0, 1 - \tau] \times [0, |\mathcal{S}|]^{\mathcal{D}}, B_0)$ for a given $\beta > 0$ such that $(1 - \delta_s(x^d))u_t^{0,d}(s, x) = \bar{u}_t^{0,d}(s, x)$ for any $t \in [0, 1 - \tau]$ and $x \in \mathcal{S}^{\mathcal{D}}$. (We allow $B_0$ to depend on $\tau$.)*

**Theorem 3** (Approximation Error). *Let $\mathcal{G}_n'$ be a class of ReLU networks with range $[m_n, M_\tau]$, depth $\mathrm{D}^* = 21(\lfloor \beta \rfloor + 1)^2 S_1 \lceil \log_2(8S_1) \rceil + 3$, width $\mathrm{W}^* = 38(\lfloor \beta \rfloor + 1)^2(\mathcal{D} + 1)^{\lfloor \beta \rfloor + 1} S_2 \lceil \log_2(8S_2) \rceil$, $S_1, S_2 \in \mathbb{N}^+$, where $m_n = |\mathcal{S}|^\beta B_0(\lfloor \beta \rfloor + 1)^2(\mathcal{D} + 1)^{\lfloor \beta \rfloor + (\beta \vee 1)/2}(S_1 S_2)^{-2\beta/(\mathcal{D}+1)}$. If Assumption 1 holds, then the approximation error has the following upper bound:*

$$\inf_{u \in \mathcal{G}_n} \mathbb{E}\Big[ \sum_{z \neq X(\mathbf{t})} D_F(u_\mathbf{t}^0(z, X(\mathbf{t})) \| u_\mathbf{t}(z, X(\mathbf{t}))) \Big]$$

$$\leq C\mathcal{D}|\mathcal{S}|^{\beta+1} B_0(\lfloor \beta \rfloor + 1)^2(\mathcal{D} + 1)^{\lfloor \beta \rfloor + \frac{\beta \vee 1}{2}}(S_1 S_2)^{-\frac{2\beta}{\mathcal{D}+1}}.$$

**Early Stopping Error Bound.** We present the early stopping error bound for the total variation between $p_1$ and $p_{1-\tau}$ in the following theorem.

**Theorem 4** (Early Stopping Error). *Consider the conditional probability path in Equation 3. The total variation between $p_1$ and $p_{1-\tau}$ has the following error bound*

$$TV(p_1, p_{1-\tau})$$

$$\leq \begin{cases} 1 - \exp\Big\{ \mathcal{D} \log \Big( -(1 - \kappa_{1-\tau}) \frac{|\mathcal{S}|-1}{|\mathcal{S}|} + 1 \Big) \Big\} & \text{if } p_0^d \sim \mathcal{U}(\mathcal{S}) \\ 1 - \exp\Big\{ \mathcal{D} \log \kappa_{1-\tau} \Big\} & \text{if } p_0^d \sim \delta_{\mathrm{m}}. \end{cases}$$

If the time schedule we used is the linear schedule $\kappa_t = t$, then the early stopping error of the discrete flow in Theorem 4 has the same convergence rate (of $\tau\mathcal{D}$) as that of the discrete diffusion derived in Theorem 1 of (Zhang et al., 2025) as $\tau \to 0^+$. In our results, we consider both masked and uniform source distribution.

**Overall Distribution Error.** Combining stochastic error, approximation error and early stopping error bounds, we are ready to derive an overall error bound for the estimated distribution.

**Theorem 5** (Distribution Error). *Suppose that we choose a linear schedule $\kappa_t = t$ and $B_0 \leq C\tau^{-1}$. Under Assumption 1, there exists a sequence of ReLU function classes $\mathcal{G}_n'$ such that for sufficiently large $n\tau^2$ the total variation between the data distribution and the estimated distribution has the following error bound (up to a logarithmic factor of $n\tau^2$):*

$$\mathbb{E}_{\mathbb{D}_n}[TV(p_1, \hat{p}_{1-\tau})]$$

$$\leq C|\mathcal{S}|^{\beta+1}\mathcal{D}^{\lfloor \beta \rfloor + 1}\tau^{-(1/2)}(n\tau^2)^{-\frac{\beta}{2(2\beta+\mathcal{D}+1)}} + C\tau\mathcal{D}.$$

**Remark 2** (Discussion on the Error Bounds). *Theorem 2 implies that the stochastic error of the ERM estimator scales with several components including covering entropy of the function class, dimension, early stopping and clipping parameters. For architecture selection, the proof of Theorem 5 suggests choosing a network class with a width not diverging with the sample size and a depth diverging with $n$. For early stopping and clipping parameter selection, combining the early stopping error, the theory suggests choosing the clipping parameter $m_n$ to have the order of $O(\tau^2)$ (if we have a fixed $\mathcal{D}|\mathcal{S}|$), since we choose $m_n$ to be our approximation error in Theorem 3.*

### 3.4. Error Analysis with Boundedness Condition

In this subsection, we derive the non-asymptotic error bound for the estimated distribution with the condition on the lower-boundedness of the oracle transition rate. We will show that the convergence rate of the estimated distribution is nearly optimal in terms of the sample size. The main difference of the results in the current and previous subsections is the error decomposition and stochastic error bounds.

**Assumption 2** (Boundedness). *The oracle rate is uniformly bounded from below: $u_t^0(z, x) > m$ uniformly for any $t \in [0, 1 - \tau], z, x \in \mathcal{S}^{\mathcal{D}}$, where $d^H(z, x) = 1$.*

**Remark 3.** *We only impose conditions on the state pairs with Hamming distance equal to one. Assumption 2 can be satisfied by choosing mixture path and uniform source distribution in Equation 3; see Section D.3 for further discussion. Assumption 2 provides the strong convexity of $F(x) = x \log x$ on $[m, M_\tau]$, and yields the bounds $|\log u^0 - \log u| \leq m^{-1}|u^0 - u|$ and $D_F(u^0 \| u) \leq m^{-1}|u^0 - u|^2$ if $u_t(z, x) \geq m$ for $d^H(z, x) = 1$. In this case, we can choose the parameter $m_n = m$ in equation 7.*

**Estimation Error Decomposition.** To utilize Assumption 2 and derive a better stochastic error bound, we consider another estimation error decomposition, which is similar to the case for $\ell_2$-loss in Jiao et al. (2023).

**Proposition 4** (Estimation Error Decomposition 2). *For random sample $\mathbb{D}_n$, the excess risk of the estimator $\hat{u}$ through*

*ERM (Equation 5) satisfies*

$$\mathbb{E}_{\mathbb{D}_n}[\mathcal{L}(\hat{u}) - \mathcal{L}(u^0)] \leq \underbrace{\mathbb{E}_{\mathbb{D}_n}[\mathcal{L}(\hat{u}) + \mathcal{L}(u^0) - 2\mathcal{L}_n(\hat{u})]}_{\textit{Stochastic Error}}$$

$$+ 2 \underbrace{\inf_{u \in \mathcal{G}_n} \mathbb{E}\Big[ \sum_{z \neq X(\mathbf{t})} D_F(u_{\mathbf{t}}^0(z, X(\mathbf{t})) \| u_{\mathbf{t}}(z, X(\mathbf{t}))) \Big]}_{\textit{Approximation Error}}.$$

**Stochastic Error Bound.** By using Assumption 2, we can derive a *fast rate* of stochastic error, which has a rate of $n^{-1}$ if we fix the function class $\mathcal{G}_n$, dimension $\mathcal{D}$ and vocabulary size $|\mathcal{S}|$.

**Theorem 6** (Stochastic Error - Fast Rate). *Assume that Assumption 2 holds. For sufficiently large $n$, the stochastic error in Proposition 4 has the following upper bound:*

$$\mathbb{E}_{\mathbb{D}_n}[\mathcal{L}(\hat{u}) + \mathcal{L}(u^0) - 2\mathcal{L}_n(\hat{u})]$$

$$\leq \frac{C(1 + \log \frac{M_\tau}{m})^2 M_\tau^2 |\mathcal{S}| \mathcal{D} \log \left[|\mathcal{S}|\mathcal{D}\mathcal{N}_n(1/(2n), \mathcal{G}_n', L^\infty)\right]}{n}.$$

**Distribution Error Bound.** The approximation error bound with the lower-boundedness condition of the oracle rate can be derived following a similar argument of Theorem 3. The following theorem shows that under the boundedness condition (Assumption 2), the distribution error has a faster convergence rate.

**Theorem 7** (Distribution Error - Fast Rate). *Suppose that we choose a linear schedule $\kappa_t = t$ and $B_0 \leq C\tau^{-1}$. Under Assumption 1 and Assumption 2, there exists a sequence of ReLU function classes $\mathcal{G}_n'$ such that for sufficiently large $n\tau^2$ the total variation between the data distribution and the estimated distribution has the following error bound (up to a logarithmic factor of $n\tau^2$):*

$$\mathbb{E}_{\mathbb{D}_n}[TV(p_1, \hat{p}_{1-\tau})]$$

$$\leq C|\mathcal{S}|^{\beta+1}\mathcal{D}^{\lfloor\beta\rfloor+\frac{3}{2}}\tau^{-1}(n\tau^2)^{-\frac{\beta}{(2\beta+\mathcal{D}+1)}} + C\tau\mathcal{D}.$$

The above distribution error bound achieves a better convergence rate, compared to the *slow rate* derived in Theorem 5. In the proof of Theorem 7, we show that the KL divergence between the estimated distribution and $p_{1-\tau}$ has a rate of $\mathcal{O}((n\tau^2)^{\frac{2\beta}{2\beta+\mathcal{D}+1}})$ (up to some logarithmic factor), which nearly matches the optimal rate of nonparametric regression for $\ell_2$-loss (Stone, 1982) for a fixed early stopping parameter $\tau$. For given $\mathcal{D}$ and $|\mathcal{S}|$, to balance the estimation error and the early stopping error, by choosing $\tau = \mathcal{O}(n^{-\frac{\beta}{6\beta+2\mathcal{D}+1}})$, the distribution error has the convergence rate $\mathcal{O}(n^{-\frac{\beta}{6\beta+2\mathcal{D}+2}})$.

**Remark 4** (Relation to Existing Works). *We derive both slow and fast convergence rates for distribution estimation in discrete flow models, parallel to the continuous flow matching results of Zhou & Liu (2025) and Gao et al.*

(2024b), respectively. Su et al. (2025) established a total variation error bound for distribution estimation with Transformer, which is of order $O(|\mathcal{S}|^{7d_0} n^{-\frac{1}{9|\mathcal{S}|d_0}} (\log n)^{\frac{1}{9|\mathcal{S}|d_0}})$ in their factorized setting; see Theorem 5.2 of Su et al. (2025). Here, they assume that the hidden dimension $d_0$ satisfies $d_0 \mid (\mathcal{D} + 1)$. However, as discussed in Section 6 of Su et al. (2025), the error bound scales with $|\mathcal{S}|^{7d_0}$, which can grow exponentially with the dimension $\mathcal{D}$. In addition, even for a fixed dimension $\mathcal{D}$, the required sample size can be exponential in the vocabulary size $|\mathcal{S}|$ in order to make the error sufficiently small. Therefore, compared with our results, their bound may not provide a meaningful guarantee for distribution estimation in discrete flow models with large vocabulary size. Moreover, our analysis explicitly accounts for the early stopping error, which is not considered in Su et al. (2025).

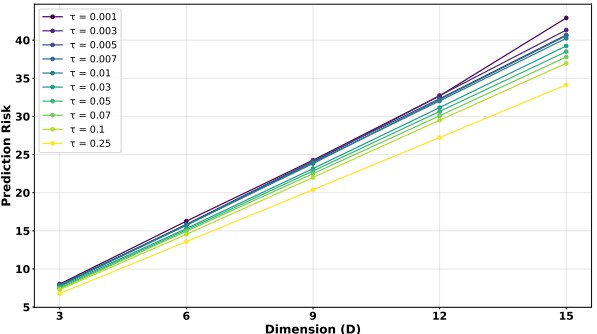

*Figure 1.* Prediction risk v.s. dimension.

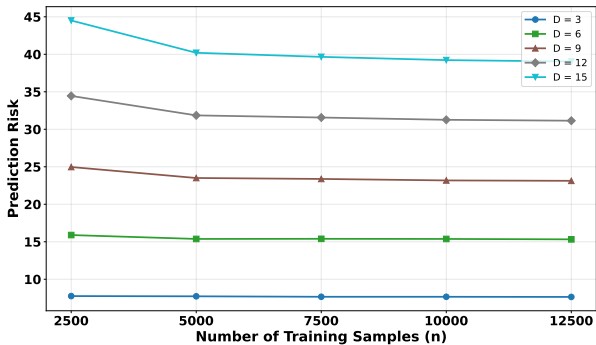

*Figure 2.* Prediction risk v.s. sample size.

## 4. Simulations

To study the empirical performance of discrete flow-based models with different dimensions, sample sizes, and hyperparameters, we conduct several simulation experiments, demonstrating the consistency of our theoretical analysis and empirical results. The data distribution and implementation details are described in Section I.1. Figure 1, Figure 2 and Figure 3 present the simulation results for the empirical version of estimation error (up to an additive constant) with

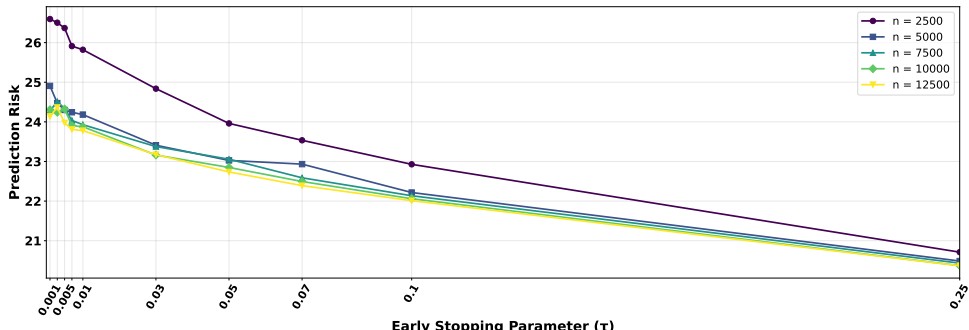

*Figure 3.* Prediction risk v.s. early stopping parameter.

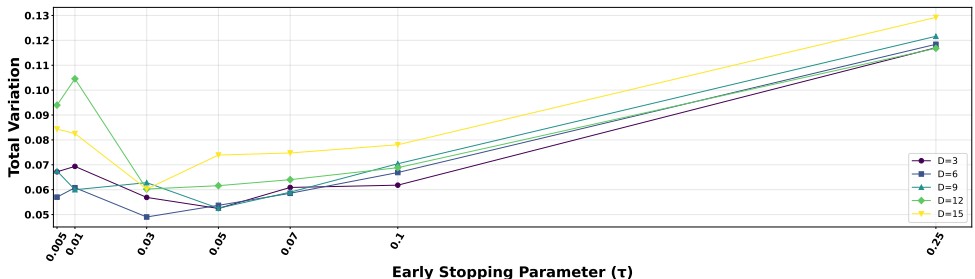

*Figure 4.* Total variation v.s. early stopping parameter with uniformization algorithm.

different sample size $n$, dimension $\mathcal{D}$ and early stopping parameter $\tau$. We can see that the estimation error is nearly linear in $\mathcal{D}$, and the estimation error decreases as $n$ and $\tau$ increase.

In addition, we also calculate the total variation (of the empirical joint distribution on the first 3 dimensions, $8^3 = 512$ states in total) with different dimension $\mathcal{D}$ and $\tau$ (see Section I.2 for details). As presented in Figure 4, the total variation decreases first and then increases as $\tau$ increases, with the minimum achieved around $\tau = 0.03$ or $0.05$.

## 5. Related Works

**Discrete Flow Models.** Discrete flow models (Campbell et al., 2024; Gat et al., 2024; Shaul et al., 2025) provide a more flexible framework to construct the transition rate and probability path than discrete diffusion models (Campbell et al., 2022; Lou et al., 2024), which also achieve superior performance in graph generation (Qin et al., 2025), visual generation and multimodal understanding (Wang et al., 2025), and video generation (Deng et al., 2025). The training objective of discrete flow models can be cross-entropy (Campbell et al., 2024; Gat et al., 2024), negative ELBO (Shaul et al., 2025), or Bregman divergence (Holderrieth et al., 2025). Su et al. (2025) established estimation error bounds for discrete flow matching using $\ell_2$-loss. In this work, we use the estimator through minimizing the empirical version of Bregman divergence (induced by

$F(x) = x \log x$) to control the distribution error.

**Error Analysis of Discrete Diffusion.** Discrete diffusion models (Austin et al., 2021; Campbell et al., 2022; Vignac et al., 2023; Sun et al., 2023; Lou et al., 2024; Benton et al., 2024b) have emerged as a CTMC-based framework for learning distribution on finite state spaces. Some recent works (Chen & Ying, 2024; Zhang et al., 2025; Pham et al., 2025; Liang et al., 2025a;b; Ren et al., 2025a;b) investigate the theoretical property of discrete diffusion models. Chen & Ying (2024) proposed to use uniformization algorithm for sampling in an exact way, and derived the distribution error bound for the scenario where the state space is a hypercube. Zhang et al. (2025) studied the theoretical results of discrete diffusion based on Girsanov-based method in the general state space $\mathcal{S}^{\mathcal{D}}$, with piece-wise homogeneous transition rates. Ren et al. (2025a) derived the error bound for both $\tau$-leaping and uniformization algorithms using stochastic integrals. (Ren et al., 2025b) proposed to use high-order $\tau$-leaping (Hu et al., 2011) for sampling and derived a better rate than Ren et al. (2025a). Liang et al. (2025a) derived the error bound for the masked source distribution. Liang et al. (2025b) improved the error bounds for $\tau$-leaping and Euler solver in terms of vocabulary size. Pham et al. (2025) established the convergence bounds under minimal data assumptions with a piece-wise constant backward rate. The training objective in these literature is the concrete score entropy introduced in Lou et al. (2024); Benton et al. (2024b). Unfortunately, in existing works, it is typical to assume strong

regularity conditions directly on the estimation error, which is related to the early stopping parameter. Furthermore, in discrete diffusion models, there is a non-zero initialization error arising from the truncation of the time horizon in the noising process.

**Error Analysis of Continuous Flow Models.** Continuous flow matching (Liu et al., 2023; Albergo & Vanden-Eijnden, 2023; Lipman et al., 2023) is a powerful simulation-free method for learning continuous data distribution based on continuous normalizing flow (Chen et al., 2018). Error bounds on the Wasserstein distance between two flow ODEs have been extensively studied (Albergo & Vanden-Eijnden, 2023; Benton et al., 2024a; Gao et al., 2024a;b; Zhou & Liu, 2025). Albergo & Vanden-Eijnden (2023) and Benton et al. (2024a) derived the Wasserstein bound in terms of the spatial Lipschitz constants of the velocity fields by utilizing Grönwall's inequality. Gao et al. (2024a) established the spatial Lipschitz regularity of the velocity field for a range of target distributions. Gao et al. (2024b); Zhou & Liu (2025) presented a comprehensive error analysis of continuous flow matching by rigorously bounding four types of errors, including early stopping error, discretization error, stochastic error and approximation error.

## 6. Conclusion and Future Works

In this work, we have established the first non-asymptotic error bounds for the total variation between the data distribution and the distribution generated by the learned transition rate in discrete flow models with a generator matching framework. We provide a general error bound for the estimated distribution without imposing strong conditions on the oracle transition rates and data distributions, by taking estimation error into account. Moreover, the distribution error bound is also derived under the boundedness condition on the oracle rate, which yields a faster convergence rate.

There are two directions deserving future research. Firstly, the error bound we derived has zero discretization error via uniformization technique; while, in practice, the uniformization algorithm does not enjoy sampling efficiency. It would be interesting to investigate the theoretical results of accuracy-efficiency trade-offs for discrete flow models. Secondly, the current work analyzes the approximation error for ReLU networks only; deriving the approximation error for networks with self-attention layers is a challenging problem in approximation theory, requiring greater effort and more careful analysis.

**Limitations.** We acknowledge two main limitations of this work.

*(1) Gap between theory and large-scale systems.* Our analysis focuses on the empirical risk minimizer (ERM) and does not consider the optimization error introduced by stochastic gradient methods in practice. Consequently, although our error bounds reveal how the sample size, dimension, vocabulary size, complexity of function class, and early stopping parameter jointly affect the convergence rate, the practical guidance they offer for hyperparameter tuning (e.g., optimal choices of $\tau$, $m_n$, network width and depth) in large-scale systems is limited. Closing this gap by jointly analyzing optimization and statistical errors is an important direction for future work.

*(2) Restricted to exact sampling via uniformization.* We adopt uniformization to obtain zero discretization error, which simplifies the analysis but is computationally expensive when the rate upper bound is large (especially as $t \to 1^-$). In practice, more efficient samplers such as $\tau$-leaping are widely used, and they can be incorporated into our framework in two possible ways. *(i)* Choose a function class with additional time-Lipschitz regularity so that the discretization error of an efficient sampler (e.g., $\tau$-leaping) can be controlled by techniques developed in concrete score matching; the corresponding approximation error analysis under this Lipschitz constraint requires more careful treatment. *(ii)* Use a masked source distribution, under which the posterior $p_{1|t}$ is time-independent (Gat et al., 2024), so that the estimated rate $\hat{u}_t(z, x) = \frac{\dot{\kappa}_t}{1 - \kappa_t}(\hat{p}_{1|t}(z|x) - \delta_x(z))$ is automatically time-Lipschitz when $\hat{p}_{1|t}$ is parameterized by a network without time embedding; the estimation error analysis can be carried out using essentially the same arguments as in this paper.

## Acknowledgements

Hongyuan Zha's research is supported in part by Shenzhen Stability Science Program 2023, and National Natural Science Foundation of China (72495131). Fang Fang gratefully acknowledges research support from National Natural Science Foundation of China (72331005). Guang Cheng gratefully acknowledges financial support through a gift from CISCO.

## Impact Statement

This paper presents work whose goal is to advance the field of machine learning. There are many potential societal consequences of our work, none of which we feel must be specifically highlighted here.

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

## Appendix

The appendix is organized as follows. Section A summarizes the main notation used throughout the paper. Section B gives the uniformization algorithm. Section C presents additional results for CTMCs. Section D offers some discussions on time singularity and conditions. Section E presents some useful lemmas. Section F gives the proof of CTMC theory. Section G and Section H provide the proof of main results. Section I presents additional simulation experiments and implementation details.

## A. Notation Table

We summarize the main notation in Table 1.

*Table 1.* Summary of key notation.

| Symbol | Description |
|---|---|
| $\mathcal{D}$ | Sequence length (dimension of the state vector) |
| $\|\mathcal{S}\|$ | Vocabulary size |
| $n$ | Sample size |
| $\tau$ | Early stopping parameter, $\tau \in (0, 1/2)$ |
| $\kappa_t$ | Time schedule, $\kappa_0 = 0, \kappa_1 = 1$ |
| $p_1$ | Target data distribution |
| $p_t$ | Marginal distribution at time $t$ generated by $u_t^0$ |
| $\hat{p}_t$ | Marginal distribution generated by the estimated rate $\hat{u}_t$ |
| $p_{t\|1}(\cdot\|x_1)$ | Conditional probability path given $X(1) = x_1$ |
| $u_t(z, x)$ | Unconditional transition rate from $x$ to $z$ at time $t$ |
| $u_t(z, x\|x_1)$ | Conditional transition rate given $X(1) = x_1$ |
| $u_t^0$ | Oracle transition rate |
| $u_t^*$ | Best approximation of $u_t^0$ in the function class $\mathcal{G}_n$ |
| $\hat{u}_t$ | ERM estimator of the transition rate, see Equation 5 |
| $M_\tau$ | Upper bound $\sup_{t\in[0,1-\tau]} \dot{\kappa}_t/(1 - \kappa_t)$ |
| $m$ | Lower bound on $u_t^0(z, x)$ for $\mathrm{d}^H(z, x) = 1$, see Assumption 2 |
| $m_n$ | Clipping parameter of the function class $\mathcal{G}_n$ |
| $\mathcal{L}(u), \mathcal{L}_n(u)$ | Population and empirical risk |
| $\mathcal{G}_n$ | Matrix-valued function class, see Equation 7 |
| $\mathcal{G}_n'$ | Scalar-valued ReLU network class used to build $\mathcal{G}_n$ |
| $\beta, B_0$ | Hölder smoothness exponent and radius, see Assumption 1 |

## B. Uniformization Algorithm

We present the uniformization algorithm for sampling in Algorithm 1.

## C. Additional Results for CTMCs

In this section, we derive the compensator of the random measure $N(t, A)$ and develop a Girsanov-type theorem for CTMCs.

---

**Algorithm 1** Sampling via Uniformization

---

**Require:** A learned transition rate $\hat{u}$, an early stopping parameter $\tau > 0$, time partition $0 = t_0 < t_1 < \cdots < t_N = 1 - \tau$, parameters $\lambda_1, \lambda_2, \ldots, \lambda_N$ satisfying $\sup_{t \in [t_k, t_{k+1}]} \sum_{z \neq x} \hat{u}_t(z, x) \leq \lambda_{k+1}$ for any $k \in \{0, 1, \ldots, N-1\}, x \in \mathcal{S}^{\mathcal{D}}$.
1: Draw $Y_0 \sim \mathcal{U}(\mathcal{S}^{\mathcal{D}})$.
2: **for** $k = 0$ to $N - 1$ **do**
3:     Draw $M \sim \text{Poisson}(\lambda_{k+1}(t_{k+1} - t_k))$.
4:     Sample $M$ points i.i.d. from $\mathcal{U}([t_k, t_{k+1}])$ and sort them as $\tau_1 < \tau_2 < \cdots < \tau_M$.
5:     Set $Z_0 = Y_k$.
6:     **for** $j = 0$ to $M - 1$ **do**
7:         Set $Z_{j+1} = \begin{cases} z, & \text{with probability } \hat{u}_{\tau_j}(z, Z_j)/\lambda_{k+1} \\ Z_j, & \text{with probability } 1 - \sum_{z \neq Z_j} \hat{u}_{\tau_j}(z, Z_j)/\lambda_{k+1} \end{cases}$, where $z \neq Z_j$.
8:     **end for**
9:     Set $Y_{k+1} = Z_M$.
10: **end for**
11: **return** $Y_N \sim \hat{p}_{1-\tau}$

---

### C.1. Compensated Random Measure

We derive the compensator of the random measure $N(t, A)$ (Definition 2) in the following proposition, where $A \subseteq \mathcal{S}^{\mathcal{D}}$.

**Proposition 5** (Compensator). *Under the assumptions in Proposition 1, the (martingale-valued) compensated random measure of $N(t, A)$ is*

$$\tilde{N}(t, A) = N(t, A) - \int_0^t \sum_{z \in A} u_s(z, X(s-)) \mathbb{1}(z \neq X(s-)) ds.$$

### C.2. Change of Measure

Suppose that $X(t)$ is a CTMC with transition rate $u^X$ and natural filtration $\mathcal{F}_t$ under probability space $(\Omega, \mathcal{F}, \mathbb{P}^X)$. Our goal is to find a probability measure $\mathbb{P}^Y \ll \mathbb{P}^X$ such that $X(t)$ is a $\mathbb{P}^Y$-CTMC with a rate $u^Y$. Denote $N(t, A)$ as the random measure associated with $X(t)$. Let $\exp(W(t))$ be an exponential $\mathbb{P}^X$-martingale (see Lemma 3 in Appendix), where $W(t)$ has the following form:

$$W(t) = W(0) + \int_0^t \sum_{x \neq X(s-)} F(s, x) ds + \int_0^t \int_{x \in \mathcal{S}^{\mathcal{D}}} K(s, x) N(ds, dx). \tag{8}$$

Since $\exp(W(t))$ is a $\mathbb{P}^X$-martingale with mean one, we can define $\frac{d\mathbb{P}^Y}{d\mathbb{P}^X}\big|_t = \frac{d\mathbb{P}^Y_t}{d\mathbb{P}^X_t} = \exp(W(t))$, where $\mathbb{P}_t$ is the restriction of the measure $\mathbb{P}$ on $(\Omega, \mathcal{F}_t)$. If we take some specific predictable processes $F$ and $K$, the following theorem shows that

$$\tilde{N}^Y(t, A) \triangleq N(t, A) - \int_0^t \sum_{z \in A} u_s^Y(z, X(s-)) \mathbb{1}(z \neq X(s-)) ds$$

is a $\mathbb{P}^Y$-martingale, and $X(t)$ is a $\mathbb{P}^Y$-CTMC with rate $u_t^Y$.

**Theorem 8** (Change of Measure). *Assume that both transition rates $u_t^X$ and $u_t^Y$ satisfy the conditions in Proposition 1. Suppose that $u_t^X(x, X(t-)) = 0$ implies $u_t^Y(x, X(t-)) = 0$. Choosing the predictable processes*

$$K(t, x) = \log \frac{u_t^Y(x, X(t-))}{u_t^X(x, X(t-))} \text{ and } F(t, x) = (u_t^X(x, X(t-)) - u_t^Y(x, X(t-))) \mathbb{1}(x \neq X(t-))$$

*in Equation 8. The R-N derivative can be written as $\frac{d\mathbb{P}^Y}{d\mathbb{P}^X}\big|_t = \exp(W(t))$, where*

$$W(t) = \int_0^t \sum_{x \neq X(s-)} [u_s^X(x, X(s-)) - u_s^Y(x, X(s-))] ds + \int_0^t \sum_{x \neq X(s-)} \log \frac{u_s^Y(x, X(s-))}{u_s^X(x, X(s-))} N(ds, x).$$

*Then $\tilde{N}^Y(t, A)$ is a $\mathbb{P}^Y$-martingale for any $A \subseteq \mathcal{S}^{\mathcal{D}}$. Moreover, $X(t)$ is a $\mathbb{P}^Y$-CTMC with rate $u_t^Y$.*

**Remark 5.** *Theorem 8 is similar to Theorem F.12 in (Pham et al., 2025), and can also be derived by applying Theorem F.12 in (Pham et al., 2025) twice, following the argument in their proof of Theorem 2.3.[1] Compared to the Girsanov's theorem used in (Zhang et al., 2025), our result can be applied to two CTMCs with arbitrary transition rates. In our result, we provide an explicit expression of the RN derivative in terms of two transition rates. This form is closely related to Theorem 9 in (Chen et al., 2023) and Theorem 5.2.12 in (Applebaum, 2009) for Brownian motion with drift.*

## D. Discussions

### D.1. Discussion on Time Singularity

We consider one token case and linear time schedule $\kappa_t = t$ for simplicity. Assume that $p_0^d$ is of uniform distribution. Note that the conditional transition rate is $u_t(z, x|x_1) = \frac{\dot{\kappa}_t}{1-\kappa_t}(\delta_{x_1}(z) - \delta_x(z))$, which can generate the conditional probability path $p_{t|1}(x|x_1) = \kappa_t \delta_{x_1}(x) + (1 - \kappa_t)p_0(x)$, where $\kappa_t \to 1$ as $t \to 1$. Then, when $p_1$ has full support, the marginal transition rate has a bounded limit as $t \to 1$:

$$
\begin{aligned}
u_t(z, x) &= \frac{\dot{\kappa}_t}{1 - \kappa_t}(p_{1|t}(z|x) - \delta_x(z)) \\
&= \frac{\dot{\kappa}_t}{1 - \kappa_t}\Big(\frac{p_1(z)p_{t|1}(x|z)}{\sum_z p_1(z)p_{t|1}(x|z)} - \delta_x(z)\Big) \\
&= \frac{\dot{\kappa}_t p_0(x)(p_1(z) - \delta_x(z))}{\kappa_t p_1(x) + (1 - \kappa_t)p_0(x)} \\
&\to \frac{\dot{\kappa}_1 p_0(x)(p_1(z) - \delta_x(z))}{p_1(x)}.
\end{aligned}
$$

When $p_1$ is not fully supported, for $x \in \{x \in \mathcal{S}^{\mathcal{D}} : p_1(x) = 0\}$, the above limit goes to infinity if $p_1(z) \in (0, 1)$ since $u_t(z, x) = \frac{\dot{\kappa}_t(p_1(z)-\delta_x(z))}{1-\kappa_t}$ in this case. This result is analogous to the continuous flow matching counterparts (see Proposition C.4 in Wan et al., 2025).

For the transition rate estimator, however, since the conditional transition rate explodes as $t \to 1$, it is hard to control the estimation error of $\hat{u}$ (see the discussion in Section D.3). Therefore, it is necessary to use the early stopping technique to balance the errors arising from estimation and early stopping.

### D.2. Discussion on $M_\tau$

We consider the following time schedules.

1. Consider the polynomial schedule $\kappa_t = t^s$ in Equation 3, where $s \geq 1$. Then, by mean value theorem, for any $z^d \neq x^d$, if $t \in (0, 1)$, we have

$$
u_t^d(z^d, x^d|x_1^d) \leq \frac{\dot{\kappa}_t}{1 - \kappa_t} = \frac{st^{s-1}}{s(t + \theta(1 - t))^{s-1}(1 - t)} \leq \frac{1}{1 - t},
$$

   which means that the conditional transition rate has the uniform upper bound $M_\tau = \frac{1}{\tau}$.

2. Consider the cosine schedule $\kappa_t = \cos^2(\frac{\pi}{2}(1 - t))$ in Equation 3, which is kinetic optimal as mentioned in (Shaul et al., 2025). Then, by $\tan(a) > a$ ($a \in (0, \pi/2)$), for any $z^d \neq x^d$, if $t \in (0, 1)$, we have

$$
u_t^d(z^d, x^d|x_1^d) \leq \frac{\dot{\kappa}_t}{1 - \kappa_t} = \frac{\pi \cos(\frac{\pi}{2}(1 - t))\sin(\frac{\pi}{2}(1 - t))}{\sin^2(\frac{\pi}{2}(1 - t))} = \frac{\pi}{\tan(\frac{\pi}{2}(1 - t))} \leq \frac{2}{1 - t},
$$

   which means that the conditional transition rate has the uniform upper bound $M_\tau = \frac{2}{\tau}$.

Given other quantities, if $\tau \to 0$, then the stochastic error will go to infinity as presented in Theorem 2 and Theorem 6, since $M_\tau = \mathcal{O}(\tau^{-1})$.

---

[1]This follows a discussion during the review process of a previous version of our paper.

### D.3. Discussion on Boundedness Condition

In this subsection, we discuss the lower bound $m$ for the oracle transition rate $u_t^0(z, x)$ (Assumption 2) in some specific scenario, which also provides a reference for time schedule selection.

We define the mixing coefficient $\alpha \in (0, 1)$ such that $\alpha \leq \frac{p_1(x_1^{\backslash d}|x_1^d)}{p_1(x_1^{\backslash d})} \leq \alpha^{-1}$ for any $x_1 \in \mathcal{S}^{\mathcal{D}}$ and $d \in [\mathcal{D}]$ (for example, if $p_1(x_1) = \prod_{d=1}^{\mathcal{D}} p_1(x_1^d)$, then $\alpha = 1$). We denote $\beta = \sup_{d, z^d, x^d}(p_1^d(z^d)/p_1^d(x^d))$, where $p_1^d$ is the marginal distribution of $p_1$ (for example, if $p_1^d$ is of uniform distribution, then $\beta = 1$). Suppose that the source distribution is uniform; that is, $p_{t|1}^d(x^d|x_1^d) = \frac{1-\kappa_t}{|\mathcal{S}|} + \kappa_t \delta_{x_1^d}(x^d)$. Then, for any $d \in [\mathcal{D}]$ and $(z, x) \in \mathcal{S}^{\mathcal{D}} \times \mathcal{S}^{\mathcal{D}}$ such that $z^d \neq x^d, z^{\backslash d} = x^{\backslash d}$, we have

$$
\begin{aligned}
u_t^0(z, x) &= \sum_{x_1} u_t^d(z^d, x^d|x_1^d) p_{1|t}(x_1|x) \\
&= \frac{\sum_{x_1} u_t^d(z^d, x^d|x_1^d) p_{t|1}^d(x^d|x_1^d) \prod_{i \neq d} p_{t|1}^i(x^i|x_1^i) p_1(x_1)}{\sum_{x_1} p_{t|1}^d(x^d|x_1^d) \prod_{i \neq d} p_{t|1}^i(x^i|x_1^i) p_1(x_1)} \\
&= \frac{\frac{\dot{\kappa}_t}{|\mathcal{S}|} p_1^d(z^d) \sum_{x_1^{\backslash d}} \prod_{i \neq d} p_{t|1}^i(x^i|x_1^i) p_1(x_1^{\backslash d}|X(1)^d = z^d)}{\frac{1-\kappa_t}{|\mathcal{S}|} \sum_{x_1^{\backslash d}} \prod_{i \neq d} p_{t|1}^i(x^i|x_1^i) p_1(x_1^{\backslash d}) + \kappa_t p_1^d(x^d) \sum_{x_1^{\backslash d}} \prod_{i \neq d} p_{t|1}^i(x^i|x_1^i) p_1(x_1^{\backslash d}|X(1)^d = x^d)} \\
&\geq \frac{|\mathcal{S}|^{-1} \dot{\kappa}_t \alpha}{(1-\kappa_t)\beta + \kappa_t \beta/\alpha} \\
&\geq |\mathcal{S}|^{-1} \dot{\kappa}_t \alpha^2/\beta.
\end{aligned}
$$

To give a specific example, we consider an AR($k$) autoregressive structure for the sequence; that is, $p_1(x_1^d|x_1^{<d}) = p_1(x_1^d|x_1^{((l-k)\vee 1):(d-1)})$ for $d > 2$. Then we have

$$
\frac{p_1(x_1^{\backslash d}|x_1^d)}{p_1(x_1^{\backslash d})}
$$

$$
= \frac{p_1(x_1^1, \ldots, x_1^{d-1}) \prod_{l=d}^{\mathcal{D}} p_1(x_1^l|x_1^{((l-k)\vee 1):(l-1)})}{p_1(x^1, \ldots, x^{d-1}) p_1(x_1^d) \prod_{j=1}^k p(x_1^{d+j}|x_1^{(((d-k)\vee 1):(d+j-1))\backslash d}) \prod_{l=d+k+1}^{\mathcal{D}} p_1(x_1^l|x_1^{((l-k)\vee 1):(l-1)})}
$$

$$
= \frac{p_1(x_1^d|x_1^{((d-k)\vee 1):(d-1)}) \prod_{j=1}^k p_1(x_1^{d+j}|x_1^{((d+j-k)\vee 1):(d+j-1)})}{p_1(x_1^d) \prod_{j=1}^k p(x_1^{d+j}|x_1^{(((d-k)\vee 1):(d+j-1))\backslash d})}.
$$

If $\log p_1(x_1^d)$ and $\log p_1(x_1^d|x_1^{((d-2k)\vee 1):(d-1)}) \in [-c, 0]$ for each $d \in [\mathcal{D}], x_1 \in \mathcal{S}^{\mathcal{D}}$, then the right-hand side of the above equation satisfying $\alpha < \frac{p_1(x_1^{\backslash d}|x_1^d)}{p_1(x_1^{\backslash d})} < \alpha^{-1}$ with $\alpha = \exp(-(k+1)c)$ and $\beta = \exp(-c)$, which are independent of $\mathcal{D}$.

Therefore, if $\alpha$ and $\beta$ are positive constants and the vocabulary size $|\mathcal{S}|$ is fixed, then $\underline{M}_c = (|\mathcal{S}|^{-1}\alpha^2/\beta) \inf_{t \in [0, 1-\tau]} \dot{\kappa}_t$ is the uniform lower bound for the oracle transition rate $u_t^0(z, x)$, where $d^H(z, x) = 1$. Empirically, one may adopt the linear schedule $\kappa_t = t$, which is commonly used in practice. More generally, a composite schedule can be employed: $\kappa_t = \frac{t + \kappa_t^0}{2}$, where $\kappa_t^0$ is a schedule that is a non-decreasing function of $t$.

## E. Some Useful Lemmas

**Lemma 1** (Exit Time). *Under the assumptions in Proposition 1, for the stopping time $T_t = \min\{h > 0 : \Delta X(t+h) \neq 0\}$, we have*

$$
\mathbb{P}(T_t > h|\mathcal{F}_t) = \exp\left(\int_t^{t+h} u_s(X(t), X(t))ds\right).
$$

*Proof.* Consider the event $E_n = \{X(t) = X(t + \frac{h}{2^n}) = \cdots = X(t+h)\}$. Since $E_{n+1} \subseteq E_n$, by dominated convergence theorem and Markov property, we have

$$
\mathbb{P}(T_t > h|\mathcal{F}_t)
$$

$$= \mathbb{P}(\cap_n E_n | \mathcal{F}_t) = \lim_{n \to \infty} \mathbb{P}(E_n | \mathcal{F}_t)$$

$$= \lim_{n \to \infty} \prod_{i=1}^{2^n} \left( \mathbb{P}\Big(X(t + \frac{i}{2^n}h) = X(t) \Big| X(t + \frac{i-1}{2^n}h) = X(t), \mathcal{F}_t \Big) \right)$$

$$= \lim_{n \to \infty} \exp\Big( \sum_{i=1}^{2^n} \frac{h}{2^n} \frac{\log \mathbb{P}\Big(X(t + \frac{i}{2^n}h) = X(t) \Big| X(t + \frac{i-1}{2^n}h) = X(t), \mathcal{F}_t \Big) - \log 1}{h/(2^n)} \Big)$$

$$= \lim_{n \to \infty} \exp\Big( \sum_{i=1}^{2^n} \frac{h}{2^n} \frac{u_{t + \frac{i-1}{2^n}h}(X(t), X(t)) \frac{h}{2^n} + R_i}{h/(2^n)} \Big)$$

$$= \exp\Big( \int_t^{t+h} u_s(X(t), X(t)) \mathrm{d}s \Big),$$

where the remainder $R_i \le C(h/2^n)^2$ for some constant $C$ not depending on $i$ by Proposition 1. $\qquad \square$

**Lemma 2** (Itô's Formula). *Consider the random measure $N(t, A)$ associated with a CTMC $X(t)$, which is defined in Definition 2. Define a stochastic integral $W(t) = W(0) + \int_0^t \int_A K(s, x) N(ds, dx)$, where $K$ is a predictable process and $A \subseteq \mathcal{S}^{\mathcal{D}}$. For each $f \in C(\mathbb{R}), t \ge 0$, we have*

$$f(W(t)) - f(W(0)) = \int_0^t \int_A [f(W(s-) + K(s, x)) - f(W(s-))] N(ds, dx).$$

*Proof.* This proof is similar to Lemma 4.4.5 in (Applebaum, 2009). Let $T_0^A = 0$ and $T_n^A = \inf\{t > T_{n-1}^A : \Delta N(t, A) \ne 0\}$. Note that

$$f(W(t)) - f(W(0)) = \sum_{n=1}^{\infty} f(W(t \wedge T_n^A)) - f(W(t \wedge T_n^A -))$$

$$= \sum_{n=1}^{\infty} [f(W(t \wedge T_n^A -) + K(t \wedge T_n^A, X(t \wedge T_n^A))) - f(W(t \wedge T_n^A -))]$$

$$= \int_0^t \int_A [f(W(s-) + K(s, x)) - f(W(s-))] N(ds, dx),$$

which completes the proof. $\qquad \square$

**Lemma 3** (Exponential Martingale). *Suppose that $N(t, A)$ is the random measure associated with the CTMC $X(t)$. Consider the following stochastic integral*

$$W(t) = W(0) + \int_0^t \sum_{x \ne X(s-)} F(s, x) ds + \int_0^t \int_{x \in \mathcal{S}^{\mathcal{D}}} K(s, x) N(ds, dx).$$

*Then $\exp(W(t))$ is an exponential martingale if for each $x \in \mathcal{S}^{\mathcal{D}}$,*

$$F(t, x) = -(e^{K(t,x)} - 1) u_t(x, X(t-)) \mathbb{1}(x \ne X(t-)).$$

*Proof.* By Itô's product formula (e.g., Theorem 4.4.13 in Applebaum, 2009) and Itô's formula (Lemma 2), we have

$$\mathrm{d}[\exp(W(t))] = \exp(W(t)) \sum_{x \ne X(t-)} F(t, x) \mathrm{d}t + \exp(W(t-)) \int_{x \in \mathcal{S}^{\mathcal{D}}} (e^{K(t,x)} - 1) N(\mathrm{d}t, \mathrm{d}x)$$

$$= \exp(W(t-)) \sum_{x \ne X(t-)} \Big\{ F(t, x) \mathrm{d}t + (e^{K(t,x)} - 1) N(\mathrm{d}t, x) \Big\}$$

$$= \exp(W(t-)) \sum_{x \ne X(t-)} \Big\{ (e^{K(t,x)} - 1) \tilde{N}(\mathrm{d}t, x) \Big\},$$

which completes the proof by Proposition 5. $\qquad \square$

**Lemma 4.** *For any random variable $Z \in \mathcal{F}_{t+h}$, we have*

$$\mathbb{E}^Y(Z|\mathcal{F}_t) = \frac{\mathbb{E}^X\big[\frac{d\mathbb{P}^Y}{d\mathbb{P}^X}\big|_{t+h} Z|\mathcal{F}_t\big]}{\frac{d\mathbb{P}^Y}{d\mathbb{P}^X}\big|_t}.$$

*Proof.* For any set $A \in \mathcal{F}_t$, we have

$$\mathbb{E}^Y(Z\mathbb{1}_A) = \mathbb{E}^X\Big[\mathbb{1}_A\mathbb{E}^X\big(\frac{d\mathbb{P}^Y}{d\mathbb{P}^X}\big|_{t+h} Z|\mathcal{F}_t\big)\Big]$$

$$= \mathbb{E}^Y\Big[\frac{\mathbb{1}_A\mathbb{E}^X(\frac{d\mathbb{P}^Y}{d\mathbb{P}^X}\big|_{t+h} Z|\mathcal{F}_t)}{\frac{d\mathbb{P}^Y}{d\mathbb{P}^X}\big|_t}\Big],$$

which completes the proof by the definition of the R-N derivative. $\square$

**Lemma 5** (Theorem 3.3 in (Jiao et al., 2023)). *Assume that $f \in \mathcal{H}^\beta([0,1]^d, B_0)$ with $\beta = s + r, s \in \mathbb{N}$ and $r \in (0,1]$. For any $S_1, S_2 \in \mathbb{N}^+$, there exists a function $\phi_0$ implemented by a ReLU network with width $\mathrm{W} = 38(\lfloor\beta\rfloor + 1)^2 d^{\lfloor\beta\rfloor+1} S_2 \lceil\log_2(8S_2)\rceil$ and depth $\mathrm{D} = 21(\lfloor\beta\rfloor + 1)^2 S_1 \lceil\log_2(8S_1)\rceil$ such that*

$$|f(x) - \phi_0(x)| \le 18B_0(\lfloor\beta\rfloor + 1)^2 d^{\lfloor\beta\rfloor+(\beta\vee1)/2}(S_1 S_2)^{-2\beta/d},$$

*for all $x \in [0,1]^d\backslash\Omega([0,1]^d, K, \delta)$, where*

$$\Omega([0,1]^d, K, \delta) = \cup_{i=1}^d\{x = [x_1, x_2, \ldots, x_d]^\top : x_i \in \cup_{k=1}^{K-1}(k/K - \delta, k/K)\},$$

*with $K = \lfloor(S_1 S_2)^{2/d}\rfloor$ and $\delta$ an arbitrary number in $(0, 1/(3K)]$*

**Lemma 6** (Theorem 12.2 in (Anthony & Bartlett, 2009)). *Let $\mathcal{G}$ be a set of real functions that map a domain $\mathcal{X}$ to a bounded interval $[0, B]$. The pseudo-dimension $Pdim(\mathcal{G})$ of $\mathcal{G}$ is defined as the largest integer $m$ for which there exists $(x_1, \ldots, x_m, y_1, \ldots, y_m) \in \mathcal{X} \times \mathbb{R}^m$ such that for any $(b_1, \ldots, b_m) \in \{0,1\}^m$ there exists $f \in \mathcal{G}$ such that $f(x_i) > y_i$ if and only if $b_i = 1$ for any $i \in [m]$. Then, for $n \ge Pdim(\mathcal{G})$ and $B \ge \epsilon$, we have*

$$\mathcal{N}_n(\epsilon, \mathcal{G}, L^\infty) \le \Big(\frac{eBn}{\epsilon Pdim(\mathcal{G})}\Big)^{Pdim(\mathcal{G})}.$$

**Lemma 7** (Theorem 7 in (Bartlett et al., 2019)). *Let $\mathcal{G}$ be a ReLU neural network function class with depth $L$ and number of parameters $S$. Then, there exists a universal constant $C$ such that*

$$Pdim(\mathcal{G}) \le CSL\log S.$$

## F. Proof of Results for CTMC

### F.1. Proof of Proposition 1

*Proof.* By construction, $X(t)$ has Markov property since the Poisson process $N(t)$ has independent increments. By conditioning argument (e.g., Theorem 3.7.9 in Durrett, 2019) and the definition of Poisson processes, for $z \ne x$, we have

$$\mathbb{P}(X(t + h) = z|X(t) = x) = Mh\exp(-Mh)\mathbb{P}(X(t + h) = z|X(t) = x, N(t + h) - N(t) = 1) + O(h^2)$$

$$= Mh\exp(-Mh) \times \Big(\int_0^h \underbrace{\frac{1}{h}}_{\text{jump location: uniform distribution}} \frac{u_{t+\theta}(z, x)}{M}d\theta\Big) + O(h^2)$$

$$= u_t(z, x)h + R_t,$$

where the remainder $R_t$ satisfying

$$R_t \le Lh^2\exp(-Mh) + \mathbb{P}(N(t + h) - N(t) > 1, X(t + h) = z|X(t) = x)$$

$$\leq \exp(-Mh)[Lh^2 + \exp(Mh) - 1 - Mh] \leq (M^2 + L)h^2 = O(h^2).$$

Consequently, since the state space is finite, we have

$$\mathbb{P}(X(t + h) = x | X(t) = x) = 1 - \sum_{z \neq x} \mathbb{P}(X(t + h) = z | X(t) = x) = 1 + u_t(x, x)h + O(h^2),$$

which completes the proof. $\square$

## F.2. Proof of Proposition 2

*Proof.* By Definition 1, we have

$$p_{t+h}(x) - p_t(x) = \sum_{z \in \mathcal{S}^{\mathcal{D}}} (p_{t+h|t}(x|z) - \delta_z(x)) p_t(z) = \sum_{z \in \mathcal{S}^{\mathcal{D}}} (u_t(x, z)h + o(h)) p_t(z).$$

Then,

$$\dot{p}_t(x) = \lim_{h \to 0^+} \frac{p_{t+h}(x) - p_t(x)}{h} = \sum_{z \in \mathcal{S}^{\mathcal{D}}} u_t(x, z) p_t(z),$$

which completes the proof. $\square$

## F.3. Proof of Proposition 5

*Proof.* In this proof, we consider the event $\{X(t) = x\}$ if we are given $\mathcal{F}_t$. Following the proof of Proposition 1, conditioning on $\mathcal{F}_t$, since the Poisson process $N(t)$ has independent increments, we have

$$\mathbb{P}(N(t + h, A) - N(t, A) = 1 | \mathcal{F}_t)$$
$$\leq \mathbb{P}(N(t + h) - N(t) = 1) \sum_{z \in A} \mathbb{P}(X(t + h) = z, X(t) \neq z | N(t + h) - N(t) = 1, \mathcal{F}_t)$$
$$+ \mathbb{P}(N(t + h) - N(t) > 1)$$
$$= \sum_{z \in A} u_t(z, X(t)) \mathbb{1}(z \neq X(t))h + R_t,$$

where $R_t$ is bounded by $|A|(M^2 + L)h^2$. Since the rate of $X(t)$ is bounded above by $M$, by uniformization, we have

$$\mathbb{E}\left[\left\{N(t + h, A) - N(t, A)\right\} \mathbb{1}(N(t + h, A) - N(t, A) \geq 2) \middle| \mathcal{F}_t\right]$$
$$\leq \mathbb{E}\left[\left\{N(t + h) - N(t)\right\} \mathbb{1}(N(t + h) - N(t) \geq 2) \middle| \mathcal{F}_t\right]$$
$$= Mh - (Mh \exp(-Mh)) \leq M^2 h^2.$$

Then,

$$\mathbb{E}\left[N(t + h, A) - N(t, A) | \mathcal{F}_t\right] = \sum_{z \in A} u_t(z, X(t)) \mathbb{1}(z \neq X(t))h + R.$$

Here, the remainder $R$ is bounded by $M^2 h^2 + |A|(M^2 + L)h^2$. Then, since $X(t)$ has only finite jumps in $[s, t]$ almost surely, by dominated convergence theorem, we have

$$\mathbb{E}\left[N(t, A) - N(s, A) \middle| \mathcal{F}_s\right]$$
$$= \sum_{i=1}^n \mathbb{E}\left[\mathbb{E}[N(s + \frac{i}{n}(t - s), A) - N(s + \frac{i-1}{n}(t - s), A) | \mathcal{F}_{s + \frac{i-1}{n}(t-s)}] \middle| \mathcal{F}_s\right]$$
$$= \sum_{i=1}^n \mathbb{E}\left[\sum_{z \in A} u_{s + \frac{i-1}{n}(t-s)}(z, X(s + \frac{i-1}{n}(t - s))) \mathbb{1}(z \neq X(s + \frac{i-1}{n}(t - s))) \frac{1}{n}(t - s) \middle| \mathcal{F}_s\right]$$

$$+ O(\frac{1}{n})$$

$$\rightarrow \mathbb{E}[\int_s^t \sum_{z \in A} u_r(z, X(r-))\mathbb{1}(z \neq X(r-))\mathrm{d}r | \mathcal{F}_s],$$

as $n \rightarrow \infty$. Thus, the process

$$\tilde{N}(t, A) = N(t, A) - \underbrace{\int_0^t \sum_{z \in A} u_s(z, X(s-))\mathbb{1}(z \neq X(s-))\,\mathrm{d}s}_{\text{predictable process}}$$

is a (martingale-valued) compensated random measure of the random measure $N(t, A)$. We are done.

$\square$

### F.4. Proof of Theorem 8

*Proof.* By computing directly, we can obtain that

$$\frac{\mathrm{d}\mathbb{P}^Y}{\mathrm{d}\mathbb{P}^X}\Big|_t = \exp\Big\{ \int_0^t \sum_{x \neq X(s-)} [u_s^X(x, X(s-)) - u_s^Y(x, X(s-))]\mathrm{d}s + \int_0^t \sum_{x \neq X(s-)} \log \frac{u_s^Y(x, X(s-))}{u_s^X(x, X(s-))} N(\mathrm{d}s, x)\Big\}.$$

Next, we divide the proof into two parts: first we prove that $\tilde{N}^Y(t, A)$ is a $\mathbb{P}^Y$-martingale, and then we conclude that $X(t)$ is a $\mathbb{P}^Y$-CTMC with rate $u_t^Y$.

$\underline{\tilde{N}^Y(t, A) \text{ is a } \mathbb{P}^Y\text{-martingale.}}$ By Itô's product formula (see Theorem 4.4.13 in Applebaum, 2009) and Lemma 3, we have

$$\mathrm{d}[\tilde{N}^Y(t, A) \exp(W(t))] = \tilde{N}^Y(t-, A) \exp(W(t-)) \sum_{x \neq X(t-)} \Big\{ (e^{K(t,x)} - 1)\tilde{N}^X(\mathrm{d}t, x) \Big\}$$

$$+ \exp(W(t-))\tilde{N}^Y(\mathrm{d}t, A)$$

$$+ \exp(W(t-)) \underbrace{\sum_{x \neq X(t-): x \in A} \Big\{ (e^{K(t,x)} - 1)N(\mathrm{d}t, x) \Big\}}_{\text{quadratic variation}}$$

$$= \tilde{N}^Y(t-, A) \exp(W(t-)) \sum_{x \neq X(t-)} \Big\{ (e^{K(t,x)} - 1)\tilde{N}^X(\mathrm{d}t, x) \Big\}$$

$$+ \exp(W(t-))\tilde{N}^X(\mathrm{d}t, A)$$

$$+ \exp(W(t-)) \sum_{x \neq X(t-): x \in A} \Big\{ (e^{K(t,x)} - 1)N(\mathrm{d}t, x) \Big\}$$

$$- \exp(W(t-)) \sum_{x \neq X(t-): x \in A} \Big[ \frac{u_t^Y(x, X(t-))}{u_t^X(x, X(t-))} - 1 \Big] u_t^X(x, X(t-))\mathrm{d}t$$

$$= \tilde{N}^Y(t-, A) \exp(W(t-)) \sum_{x \neq X(t-)} \Big\{ \frac{u_t^Y(x, X(t-))}{u_t^X(x, X(t-))} - 1)\tilde{N}^X(\mathrm{d}t, x) \Big\}$$

$$+ \exp(W(t-)) \sum_{x \neq X(t-): x \in A} \frac{u_t^Y(x, X(t-))}{u_t^X(x, X(t-))} \tilde{N}^X(\mathrm{d}t, x),$$

which implies that $\tilde{N}^Y(t, A) \exp(W(t))$ is a $\mathbb{P}^X$-martingale. By Lemma 5.2.11 in (Applebaum, 2009), $\tilde{N}^Y(t, A)$ is a $\mathbb{P}^Y$-martingale.

$\underline{X(t) \text{ is a } \mathbb{P}^Y\text{-CTMC with rate } u_t^Y.}$ Considering $Z = \mathbb{1}(X(t+h) = z)$ in Lemma 4, we can obtain that

$$\mathbb{P}^Y(X(t+h) = z | \mathcal{F}_t) = \mathbb{E}^X[\exp(W(t+h) - W(t))\mathbb{1}(X(t+h) = z)|\mathcal{F}_t].$$

Since the right-hand side in above equation only depends on $X(t)$, thus $X(t)$ has Markov property under $\mathbb{P}^Y$. It suffices to show that $\mathbb{P}^Y(X(t+h) = z|\mathcal{F}_t) = u_t(z, X(t))h + o(h)$ for any $z \neq X(t)$.

Since $\exp(W(t+h) - W(t)) \leq \exp(Ch)$ for some constant $C$, by Lemma 4, we have

$$\mathbb{P}^Y(N(t+h, A) - N(t, A) > 1|\mathcal{F}_t) = \frac{\mathbb{E}^X[\frac{d\mathbb{P}^Y}{d\mathbb{P}^X}\big|_{t+h} \mathbb{1}(N(t+h, A) - N(t, A) > 1)|\mathcal{F}_t]}{\frac{d\mathbb{P}^Y}{d\mathbb{P}^X}\big|_t}$$

$$\leq \exp(Ch)\mathbb{P}^X(N(t+h, A) - N(t, A) > 1|\mathcal{F}_t)$$

$$= O(h^2).$$

Note that, for $\mathbb{P}^Y$-submartingale $(\tilde{N}^Y)^2$, by Itô product formula, we have Doob-Meyer decomposition:

$$d(\tilde{N}^Y)^2(t, A) = 2\tilde{N}^Y(t-)d\tilde{N}^Y(t, A) + d[\tilde{N}^Y, \tilde{N}^Y](t, A)$$

$$= 2\tilde{N}^Y(t-, A)d\tilde{N}^Y(t, A) + dN(t, A).$$

Then, by Lemma 4 again, $\mathbb{E}^Y[(\tilde{N}^Y(t+h, A) - \tilde{N}^Y(t, A))^2|\mathcal{F}_t] = \mathbb{E}^Y[N(t+h, A) - N(t, A)|\mathcal{F}_t] = O(h)$. By Cauchy-Schwarz inequality, we can obtain

$$\mathbb{E}^Y[(N(t+h, A) - N(t, A))\mathbb{1}(N(t+h, A) - N(t, A) \geq 2)|\mathcal{F}_t]$$

$$\leq \sqrt{\mathbb{E}^Y\left[(N(t+h, A) - N(t, A))^2|\mathcal{F}_t\right]\mathbb{P}^Y\left(N(t+h, A) - N(t, A) > 1|\mathcal{F}_t\right)}$$

$$\leq \sqrt{\left(2\mathbb{E}^Y[(\tilde{N}^Y(t+h, A) - \tilde{N}^Y(t, A))^2|\mathcal{F}_t] + 2\mathbb{E}^Y\left[\left(\int_t^{t+h} \sum_{z \in A} u_s^Y(z, X(s-))\mathbb{1}(z \neq X(s-))ds\right)^2 \Big|\mathcal{F}_t\right]\right)}$$

$$\times \sqrt{\mathbb{P}^Y(N(t+h, A) - N(t, A) > 1|\mathcal{F}_t)}$$

$$= O(h^{3/2})$$

Then, conditioning on $\mathcal{F}_t$, for $z \neq X(t)$, we have

$$\mathbb{P}^Y(X(t+h) = z|\mathcal{F}_t) \leq \mathbb{P}^Y(N(t+h, z) - N(t, z) = 1|\mathcal{F}_t) + O(h^2)$$

$$= \mathbb{E}^Y(N(t+h, z) - N(t, z)|\mathcal{F}_t) + O(h^{3/2})$$

$$= \int_t^{t+h} u_t^Y(z, X(t))ds + \int_t^{t+h} u_s^Y(z, X(t)) - u_t^Y(z, X(t))ds$$

$$+ \mathbb{E}^Y(\int_t^{t+h} u_s^Y(z, X(s-)) - u_s^Y(z, X(t))ds|\mathcal{F}_t) + O(h^{3/2})$$

$$= u_t(z, X(t))h + O(h^{3/2}),$$

since

$$\mathbb{E}^Y(\int_t^{t+h} u_s^Y(z, X(s-)) - u_s^Y(z, X(t))ds|\mathcal{F}_t) \leq 2Mh\mathbb{P}^Y(N(t+h, \mathcal{S}^{\mathcal{D}}) - N(t, \mathcal{S}^{\mathcal{D}}) \geq 1|\mathcal{F}_t)$$

$$\leq 2Mh\mathbb{E}^Y(N(t+h, \mathcal{S}^{\mathcal{D}}) - N(t, \mathcal{S}^{\mathcal{D}})|\mathcal{F}_t)$$

$$\leq 2M^2h^2 = O(h^2).$$

This completes the proof.

$\square$

### F.5. Proof of Theorem 1

*Proof.* By Theorem 8, we compute directly:

$$D_{\text{KL}}(\mathbb{P}_t^X||\mathbb{P}_t^Y) = \mathbb{E}^X\left[\log(\frac{d\mathbb{P}^X}{d\mathbb{P}^Y}\big|_t)\right]$$

$$= -\mathbb{E}^X \Big\{ \int_0^t \sum_{x \neq X(s-)} [u_s^X(x, X(s-)) - u_s^Y(x, X(s-))] + u_s^X(x, X(s-)) \log \frac{u_s^Y(x, X(s-))}{u_s^X(x, X(s-))} \mathrm{d}s \Big\}$$

$$- \mathbb{E}^X \Big\{ \int_0^t \sum_{x \neq X(s-)} \log \frac{u_s^Y(x, X(s-))}{u_s^X(x, X(s-))} \tilde{N}^X(\mathrm{d}s, x) \Big\}$$

$$= \mathbb{E}^X \Big\{ \int_0^t \sum_{x \neq X(s)} D_F(u_s^X(x, X(s)) \| u_s^Y(x, X(s))) \mathrm{d}s \Big\},$$

where the last equation holds since the integrator is the Lebesgue measure.

By Jensen's inequality, we have

$$D_{\mathrm{KL}}(p_t^X \| p_t^Y) = \mathbb{E}^X \Big[ \log(\frac{\mathrm{d}\mathbb{P}^X}{\mathrm{d}\mathbb{P}^Y}\big|_{\sigma(X_t)}) \Big]$$

$$\leq \mathbb{E}^X \Big[ \log(\frac{\mathrm{d}\mathbb{P}^X}{\mathrm{d}\mathbb{P}^Y}\big|_{\mathcal{F}_t}) \Big]$$

$$= \mathbb{E}^X \Big\{ \int_0^t \sum_{x \neq X(s)} D_F(u_s^X(x, X(s)) \| u_s^Y(x, X(s))) \mathrm{d}s \Big\}.$$

This completes the proof. $\qquad\square$

## G. Proof of Main Results in Section 3.3

### G.1. Proof of Proposition 3

*Proof.* Using the definition of empirical risk minimization, we have $\mathcal{L}_n(\hat{u}) \leq \mathcal{L}_n(u^*)$. Thus, it holds that

$$\mathbb{E}[\mathcal{L}(\hat{u}) - \mathcal{L}(u^0)] \leq \mathbb{E}\Big[ \mathcal{L}(\hat{u}) - \mathcal{L}_n(\hat{u}) + \mathcal{L}_n(\hat{u}) - \mathcal{L}_n(u^*) + \mathcal{L}_n(u^*) - \mathcal{L}(u^0) \Big]$$

$$\leq \mathbb{E} \sup_{u \in \mathcal{G}_n} \Big| \mathcal{L}_n(u) - \mathcal{L}(u) \Big| + \mathcal{L}(u^*) - \mathcal{L}(u^0),$$

which completes the proof by using Equation 6. $\qquad\square$

### G.2. Proof of Theorem 2

*Proof.* Note that

$$\mathbb{E} \sup_{u \in \mathcal{G}_n} \Big| \mathcal{L}_n(u) - \mathcal{L}(u) \Big| \leq \inf_{\eta > 0} \Big\{ \eta + \int_\eta^\infty \mathbb{P}\Big( \exists u \in \mathcal{G}_n, \Big| \mathcal{L}_n(u) - \mathcal{L}(u) \Big| \geq t \Big) \Big\}.$$

Next, we aim to bound the above tail probability. Recall that $u_t(z, x) = \sum_{d \in [\mathcal{D}]} \delta_{x^{\backslash d}}(z^{\backslash d}) u_t^d(z^d, x)$ for any $z \neq x$. Therefore, we can focus only on $u_t^d(s, x)$ as a function of $(x, t)$, given $(d, s) \in \mathcal{D} \times \mathcal{S}$. Thus, the loss function can be rewritten as

$$\mathcal{L}_n(u) = \frac{1}{n} \sum_{i=1}^n \sum_{d \in [\mathcal{D}]} \sum_{s \in \mathcal{S}} \Big\{ \Big( 1 - \delta_{X_i(\mathbf{t})^d}(s) \Big) \Big[ -u_{\mathbf{t}}^d(s, X_i(\mathbf{t})^d | X_i(1)^d) \log u_{\mathbf{t}}^d(s, X_i(\mathbf{t})) + u_{\mathbf{t}}^d(s, X_i(\mathbf{t})) \Big] \Big\}$$

$$\triangleq \frac{1}{n} \sum_{i=1}^n \sum_{d \in [\mathcal{D}]} \sum_{s \in \mathcal{S}} \mathcal{J}(u, d, s, Z_i).$$

Similarly, we have

$$\mathcal{L}(u) = \sum_{d \in [\mathcal{D}]} \sum_{s \in \mathcal{S}} \mathbb{E}[\mathcal{J}(u, d, s, Z)].$$

By symmetrization argument for probabilities (e.g. Lemma 2.3.7 in van der Vaart & Wellner, 2023), we have

$$\mathbb{P}\Big(\exists u \in \mathcal{G}_n, \big|\mathcal{L}_n(u) - \mathcal{L}(u)\big| \geq t\Big)$$

$$\leq \mathbb{P}\Big(\exists(u,d,s) \in \mathcal{G}_n \times [\mathcal{D}] \times \mathcal{S}, \Big|\frac{1}{n}\sum_{i=1}^{n}\mathcal{J}(u,d,s,Z_i) - \mathbb{E}[\mathcal{J}(u,d,s,Z)]\Big| \geq \frac{t}{\mathcal{D}|\mathcal{S}|}\Big)$$

$$\leq \frac{2}{\beta_n(\frac{tn}{\mathcal{D}|\mathcal{S}|})}\mathbb{P}\Big(\exists(u,d,s) \in \mathcal{G}_n \times [\mathcal{D}] \times \mathcal{S}, \Big|\frac{1}{n}\sum_{i=1}^{n}\epsilon_i\Big(\mathcal{J}(u,d,s,Z_i) - \mathbb{E}[\mathcal{J}(u,d,s,Z)]\Big)\Big| \geq \frac{t}{4\mathcal{D}|\mathcal{S}|}\Big),$$

where $\{\epsilon_i\}_{i=1}^{n}$ is a sequence of i.i.d. Rademacher variables, and

$$\beta_n\Big(\frac{tn}{\mathcal{D}|\mathcal{S}|}\Big) \geq 1 - \Big(\frac{4\mathcal{D}^2|\mathcal{S}|^2}{nt^2}\Big)\sup_{(u,d,s)\in(\mathcal{G}_n,\mathcal{D},\mathcal{S})}\mathbb{E}[\mathcal{J}^2(u,d,s,Z)] \geq 1 - \Big(\frac{4\mathcal{D}^2|\mathcal{S}|^2}{nt^2}\Big)M^2(\log m_n^{-1}+1)^2. \tag{9}$$

Given $\mathbb{D}_n$, we consider the following sequence with length $n$:

$$\mathbf{s}_n = \Big\{(\mathbf{t}_i, X_i(\mathbf{t}_i))\Big\}_{i\in[n]},$$

Let $\mathcal{H}_\delta^{d,s}(\mathbb{D}_n)$ be a $L^\infty$ $\delta$-covering of $\mathcal{G}_n'|_{\mathbf{s}_n}$ with minimal size. That is to say, given $\mathbb{D}_n$, $d \in [\mathcal{D}]$ and $s \in \mathcal{S}$, for any $u_\cdot^d(s,\cdot) \in \mathcal{G}_n'$, there exists $\tilde{u}_\cdot^d(s,\cdot) \in \mathcal{H}_\delta^{d,s}(\mathbb{D}_n)$ such that for any $i \in [n]$

$$|u_{\mathbf{t}_i}^d(s, X_i(\mathbf{t}_i)) - \tilde{u}_{\mathbf{t}_i}^d(s, X_i(\mathbf{t}_i))| \leq \delta,$$

which implies that

$$|\mathcal{J}(u,d,s,Z_i) - \mathcal{J}(\tilde{u},d,s,Z_i)| \leq \Big(\frac{M_\tau}{m_n} + 1\Big)\delta. \tag{10}$$

Define

$$\mathcal{H}_\delta(\mathbb{D}_n) = \Big\{\{u_\cdot^d(s,\cdot)\}_{d\in\mathcal{D},s\in\mathcal{S}} : \mathcal{S}^{\mathcal{D}} \times [0,1] \to \mathbb{R}^{\mathcal{D}\times|\mathcal{S}|} : |u_t^d(z^d,x)| \geq m_n, u_\cdot^d(s,\cdot) \in \mathcal{H}_\delta^{d,s}(\mathbb{D}_n)\Big\}.$$

By Hoffeding's inequality (e.g., Theorem 2.6.2 of Vershynin, 2018), conditioning argument and union bound, if $\delta \leq \frac{t}{16(\frac{M}{m_n}+1)\mathcal{D}|\mathcal{S}|}$, then we have

$$\mathbb{P}\Big(\exists(u,d,s) \in \mathcal{G}_n \times [\mathcal{D}] \times \mathcal{S}, \Big|\frac{1}{n}\sum_{i=1}^{n}\epsilon_i\Big(\mathcal{J}(u,d,s,Z_i) - \mathbb{E}[\mathcal{J}(u,d,s,Z)]\Big)\Big| \geq \frac{t}{4\mathcal{D}|\mathcal{S}|}\Big)$$

$$\leq \mathbb{P}\Big\{\exists(u,d,s) \in \mathcal{H}_n(\mathbb{D}_n) \times [\mathcal{D}] \times \mathcal{S}, \Big|\frac{1}{n}\sum_{i=1}^{n}\epsilon_i\Big(\mathcal{J}(u,d,s,Z_i) - \mathbb{E}[\mathcal{J}(u,d,s,Z)]\Big)\Big| \geq \frac{t}{8\mathcal{D}|\mathcal{S}|}\Big\}$$

$$\leq 2\mathbb{E}\Big[\mathcal{D}|\mathcal{S}|\mathcal{N}_n(\delta, \mathcal{G}_n'|_{\mathbf{s}_n}, L^\infty)\exp\Big(-\frac{Cnt^2}{M_\tau^2(\log m_n^{-1}+1)^2\mathcal{D}^2|\mathcal{S}|^2}\Big)\Big]$$

$$\leq 2\mathcal{D}|\mathcal{S}|\mathcal{N}_n(\delta, \mathcal{G}_n', L^\infty)\exp\Big(-\frac{Cnt^2}{M_\tau^2(\log m_n^{-1}+1)^2\mathcal{D}^2|\mathcal{S}|^2}\Big). \tag{11}$$

Choosing

$$\eta_n^* = CM_\tau(\log m_n^{-1}+1)\mathcal{D}|\mathcal{S}|\sqrt{\frac{\log[\mathcal{D}|\mathcal{S}|\mathcal{N}_n(\delta,\mathcal{G}_n',L^\infty)]}{n}},$$

we have

$$\mathbb{E}\sup_{u\in\mathcal{G}_n}\big|\mathcal{L}_n(u) - \mathcal{L}(u)\big| \leq \Big\{\eta_n^* + \int_{\eta_n^*}^{\infty}\mathbb{P}\Big(\exists u \in \mathcal{G}_n, \big|\mathcal{L}_n(u) - \mathcal{L}(u)\big| \geq t\Big)\Big\}$$

$$\leq \eta_n^* + \frac{CM_\tau^2(\log m_n^{-1} + 1)^2 \mathcal{D}^3 |\mathcal{S}|^3 \mathcal{N}_n(\delta, \mathcal{G}_n', L^\infty)}{n\eta_n^*} \exp\Big(-\frac{Cn(\eta_n^*)^2}{M_\tau^2(\log m_n^{-1} + 1)^2 \mathcal{D}^2 |\mathcal{S}|^2}\Big)$$

$$\leq CM_\tau(\log m_n^{-1} + 1)\mathcal{D}|\mathcal{S}|\sqrt{\frac{\log[\mathcal{D}|\mathcal{S}|\mathcal{N}_n(\delta, \mathcal{G}_n', L^\infty)]}{n}},,$$

provided that $\delta \leq \frac{\eta_n^*}{16(\frac{M_\tau}{m_n}+1)\mathcal{D}|\mathcal{S}|}$. This completes the proof by choosing $\delta = cm_n/(\sqrt{n})$.

$\square$

### G.3. Proof of Theorem 3

*Proof.* We prove that there exists a sequence of neural networks with the ReLU activation function that can control the approximation error. We focus on the region $(t, x) \in [0,1] \times [0, |\mathcal{S}|]^{\mathcal{D}}$. For any $d \in \mathcal{D}$ and $s \in \mathcal{S}$, if we can find a network $u_\cdot^{1,d}(s, \cdot) : [0,1] \times [0, |\mathcal{S}|]^{\mathcal{D}} \to [m_n, B_0]$ such that $|(1 - \delta_s(x^d))u_t^{0,d}(s, x) - u_t^{1,d}(s, x)| < \epsilon$ for any $(t, x) \in [0, 1 - \tau] \times [0, |\mathcal{S}|]^{\mathcal{D}}$, then the approximation error can be bounded by $\frac{1}{m_n}\mathcal{D}|\mathcal{S}|(\epsilon \vee m_n)^2$. It suffices to analyze the approximation error for a given $d \in [D]$ and $s \in \mathcal{S}$.

Let $f^{0,d,s}(t, x) = (1 - \delta_s(x^d))u_t^{0,d}(s, x)$ and $\tilde{u}_t^{0,d}(s, x') = f^{0,d,s}(t, x'|\mathcal{S}|)$ for $x' \in [0,1]^{\mathcal{D}}$. Then

$$\tilde{u}^{0,d}(s, \cdot) \in \mathcal{H}^\beta([0,1]^{\mathcal{D}+1}, |\mathcal{S}|^\beta M_\tau).$$

Lemma 5 implies that for any $S_1, S_2 \in \mathbb{N}^+$, there exists a function $f^*$ implemented by a ReLU network with depth $D = 21(\lfloor\beta\rfloor + 1)^2 S_1 \lceil\log_2(8S_1)\rceil$ and width $W = 38(\lfloor\beta\rfloor + 1)^2(\mathcal{D} + 1)^{\lfloor\beta\rfloor+1} S_2 \lceil\log_2(8S_2)\rceil$ such that

$$|f^*(d, s, t, x') - \tilde{u}_t^{0,d}(s, x')| \leq 18|\mathcal{S}|^\beta B_0(\lfloor\beta\rfloor + 1)^2(\mathcal{D} + 1)^{\lfloor\beta\rfloor+(\beta\vee 1)/2}(S_1 S_2)^{-2\beta/(\mathcal{D}+1)},$$

for any $(t, x') \in [0,1]^{\mathcal{D}+1}\backslash\Omega([0,1]^{\mathcal{D}+1}, K, \delta)$, where $K = \lceil(S_1 S_2)^{2/(\mathcal{D}+1)}\rceil$ and $\delta$ is an arbitrary number in $(0, 1/(3K))$. Let $f^{**}(d, s, t, x) = f^*(d, s, t, x/|\mathcal{S}|)$ for $x \in [0, |\mathcal{S}|]^{\mathcal{D}}$ be a network with depth $D + 1$. Note that a clip function can be expressed as a two-layer ReLU network, then we define the NN $f^{***}(d, s, t, x) = m_n + \text{ReLU}(M_\tau - \text{ReLU}(M_\tau - f^{**}(d, s, t, x)) - m_n)$ with depth $D + 3$, whose range is $[m_n, M_\tau]$.

Consequently, we have the following bound

$$|f^{***}(d, s, t, x) - f^{0,d,s}(t, x)| \leq 18|\mathcal{S}|^\beta B_0(\lfloor\beta\rfloor + 1)^2(\mathcal{D} + 1)^{\lfloor\beta\rfloor+(\beta\vee 1)/2}(S_1 S_2)^{-2\beta/(\mathcal{D}+1)} \vee m_n,$$

for any $(t, x) \in [0, 1] \times [0, |\mathcal{S}|]^{\mathcal{D}}$ such that $(t, x/|\mathcal{S}|) \in [0,1]^{\mathcal{D}+1}\backslash\Omega([0,1]^{\mathcal{D}+1}, K, \delta)$, where

$$\Omega([0,1]^{\mathcal{D}+1}, K, \delta) = \cup_{i=1}^{\mathcal{D}+1}\{x = [x_1, x_2, \ldots, x_{\mathcal{D}+1}]^\top : x_i \in \cup_{k=1}^{K-1}(k/K - \delta, k/K)\}.$$

Here, $K = \lfloor(S_1 S_2)^{2/(\mathcal{D}+1)}\rfloor$ and $\delta$ is an arbitrary number in $(0, 1/(3K)]$. Note that $\mathbf{t} \sim \mathcal{U}([0, 1 - \tau])$, whose probability measure is absolutely continuous w.r.t. the Lebesgue measure. Moreover, when $\delta$ is sufficiently small, the set $\{1/|\mathcal{S}|, 2/|\mathcal{S}|, \ldots, 1\}$ and the set $\cup_{k=1}^{K-1}(k/K - \delta, k/K)$ are disjoint. Then, the event

$$\{(\mathbf{t}, x/|\mathcal{S}|) \in [0,1]^{\mathcal{D}+1}\backslash\Omega([0,1]^{\mathcal{D}+1}, K, \delta) \text{ for any } x \in \mathcal{S}^{\mathcal{D}}\}$$

holds with probability greater than $1 - 2K\delta$. Thus, since $\delta$ is an arbitrary number in $(0, 1/(3K))$, by choosing $m_n = |\mathcal{S}|^\beta B_0(\lfloor\beta\rfloor + 1)^2(\mathcal{D} + 1)^{\lfloor\beta\rfloor+(\beta\vee 1)/2}(S_1 S_2)^{-2\beta/(\mathcal{D}+1)}$, the approximation error has the following bound (we use $D_F(a||b) = b[(a/b)\log(a/b) - (a/b) + 1] \leq b[(a/b) - 1]^2 \leq (a - b)^2/b$ for any $a \geq 0, b > 0$)

$$\inf_{u\in\mathcal{G}_n} \mathbb{E}\Big[\sum_{z\neq X(\mathbf{t})} D_F\Big(u_\mathbf{t}^0(z, X(\mathbf{t}))||u_\mathbf{t}(z, X(\mathbf{t}))\Big)\Big]$$

$$\leq \frac{1}{m_n}\mathbb{E}\Big\{\sum_{d\in[\mathcal{D}],s\in\mathcal{S}} \Big|f^{***}(d, s, \mathbf{t}, X(\mathbf{t})) - f^{0,d,s}(\mathbf{t}, X(\mathbf{t}))\Big|^2\Big\} \tag{12}$$

$$\leq C\mathcal{D}|\mathcal{S}|^{\beta+1} B_0(\lfloor\beta\rfloor + 1)^2(\mathcal{D} + 1)^{\lfloor\beta\rfloor+(\beta\vee 1)/2}(S_1 S_2)^{-2\beta/(\mathcal{D}+1)},$$

where $\mathcal{G}_n$ is the class of matrix-valued ReLU networks with range $[m_n, M_\tau]^{\mathcal{D}\times|\mathcal{S}|}$, depth $D^* = 21(\lfloor\beta\rfloor+1)^2 S_1 \lceil\log_2(8S_1)\rceil + 3$, width $W^* = 38(\lfloor\beta\rfloor + 1)^2(\mathcal{D} + 1)^{\lfloor\beta\rfloor+1} S_2 \lceil\log_2(8S_2)\rceil$, $S_1, S_2 \in \mathbb{N}^+$.

$\square$

## G.4. Proof of Theorem 4

*Proof.* **Uniform source distribution.** We follow the proof of Theorem 6 (2) in (Chen & Ying, 2024) and Theorem 1 in (Zhang et al., 2025). We have

$$
\begin{aligned}
\mathrm{TV}(p_1, p_{1-\tau}) &\leq \mathbb{P}(X(1) \neq X(1 - \tau)) \\
&= 1 - \sum_{x_1} p(x_1) \prod_{d=1}^{\mathcal{D}} p_{1-\tau|1}^d(x_1^d | x_1^d) \\
&= 1 - \left( \kappa_{1-\tau} + \frac{1 - \kappa_{1-\tau}}{|\mathcal{S}|} \right)^{\mathcal{D}} \\
&= 1 - \exp\left\{ \mathcal{D} \log\left( -(1 - \kappa_{1-\tau}) \frac{|\mathcal{S}| - 1}{|\mathcal{S}|} + 1 \right) \right\}.
\end{aligned}
$$

**Masked source distribution.** Using a similar argument, we have

$$
\begin{aligned}
\mathrm{TV}(p_1, p_{1-\tau}) &\leq \mathbb{P}(X(1) \neq X(1 - \tau)) \\
&= 1 - \sum_{x_1} p(x_1) \prod_{d=1}^{\mathcal{D}} p_{1-\tau|1}^d(x_1^d | x_1^d) \\
&= 1 - \kappa_{1-\tau}^{\mathcal{D}} \\
&= 1 - \exp\left\{ \mathcal{D} \log \kappa_{1-\tau} \right\},
\end{aligned}
$$

which completes the proof. $\qquad\square$

## G.5. Proof of Theorem 5

*Proof.* In this proof, the error bound might omit a logarithmic multiplier of $n\tau^2$.

**Step 1. Choosing hyperparameters.** By Theorem 2, Lemma 6 and Lemma 7, the stochastic error is bounded by

$$
\mathbb{E} \sup_{u \in \mathcal{G}_n} \left| \mathcal{L}_n(u) - \mathcal{L}(u) \right| \leq C |\mathcal{S}| \mathcal{D} \log(n m_n^{-1}) \sqrt{\frac{\mathrm{S}^* \mathrm{D}^* \log \mathrm{S}^*}{n \tau^2}},
$$

where $\mathrm{S}^*$ is the number of parameters of the ReLU networks in $\mathcal{G}_n'$. Note that for a ReLU network with depth $\mathrm{D}^*$ and width $\mathrm{W}^*$ and input dimension $\mathcal{D} + 1$, we have (assume that $\mathcal{D} \lesssim \mathrm{W}^* \mathrm{D}^*$)

$$
\mathrm{S}^* \leq \underbrace{\mathrm{W}^*(\mathcal{D} + 1) + \mathrm{W}^*}_{\text{input layer}} + \underbrace{((\mathrm{W}^*)^2 + \mathrm{W}^*)(\mathrm{D}^* - 1)}_{\text{hidden layer}} + \underbrace{\mathrm{W}^* + 1}_{\text{output layer}} = O((\mathrm{W}^*)^2 \mathrm{D}^*).
$$

Therefore, by Theorem 3, choosing $S_1 = C(n\tau^2)^{\frac{\mathcal{D}+1}{(2\mathcal{D}+4\beta+2)}}$ and $S_2 = C$, we have

$$
\mathrm{W}^* = C(\mathcal{D} + 1)^{\lfloor \beta \rfloor + 1}; \quad \mathrm{D}^* = C(n\tau^2)^{\frac{\mathcal{D}+1}{(2\mathcal{D}+4\beta+2)}} \log(n\tau^2); \quad \mathrm{S}^* = C\mathcal{D}^{2\lfloor \beta \rfloor + 2}(n\tau^2)^{\frac{\mathcal{D}+1}{(2\mathcal{D}+4\beta+2)}} \log(n\tau^2),
$$

yielding that

$$
\mathbb{E} \sup_{u \in \mathcal{G}_n} \left| \mathcal{L}_n(u) - \mathcal{L}(u) \right| \leq C |\mathcal{S}| \mathcal{D}^{\lfloor \beta \rfloor + 2} (n\tau^2)^{-\frac{2\beta}{(2\mathcal{D}+4\beta+2)}} \log(n\tau^2) \log(n m_n^{-1}),
$$

if $n \geq \mathcal{D}$.

By Theorem 3, the approximation error is

$$
\inf_{u \in \mathcal{G}_n} \mathbb{E}\left[ D_F\left( \sum_{z \neq X(\mathbf{t})} u^0(\mathbf{t}, z, X(\mathbf{t})) \,\|\, \sum_{z \neq X(\mathbf{t})} u(\mathbf{t}, z, X(\mathbf{t})) \right) \right] \leq C |\mathcal{S}|^{\beta+1} \mathcal{D}^{2\lfloor \beta \rfloor + 2} \tau^{-1} (n\tau^2)^{-\frac{2\beta}{(2\mathcal{D}+4\beta+2)}}
$$

Then the summation of the approximation error and the stochastic error has the convergence rate (up to some logarithmic multiplier)

$$\mathbb{E}_{\mathbb{D}_n}[D_{KL}(p_{1-\tau}||\hat{p}_{1-\tau})] \leq C|\mathcal{S}|^{\beta+1}\mathcal{D}^{2\lfloor\beta\rfloor+2}\tau^{-1}(n\tau^2)^{-\frac{2\beta}{(2\mathcal{D}+4\beta+2)}}. \tag{13}$$

**Step 2. Deriving final results.** By triangle inequality, Pinsker's inequality and Theorems 2 and 3, for sufficiently large $n\tau^2$, we have

$$\begin{aligned}
\mathbb{E}_{\mathbb{D}_n}\Big[\text{TV}(p_1, \hat{p}_{1-\tau})\Big] &\leq \mathbb{E}_{\mathbb{D}_n}\Big[\text{TV}(p_{1-\tau}, \hat{p}_{1-\tau})\Big] + \text{TV}(p_1, p_{1-\tau}) \\
&\leq \mathbb{E}_{\mathbb{D}_n}[D_{KL}(p_{1-\tau}||\hat{p}_{1-\tau})] + \text{TV}(p_1, p_{1-\tau}) \\
&\leq C|\mathcal{S}|^{\beta+1}\mathcal{D}^{\lfloor\beta\rfloor+1}\tau^{-(1/2)}(n\tau^2)^{-\frac{\beta}{(2\mathcal{D}+4\beta+2)}} + C\tau\mathcal{D},
\end{aligned}$$

which completes the proof.

$\square$

# H. Proof of Results in Section 3.4

## H.1. Proof of Proposition 4

*Proof.* This proof is similar to the proof of Lemma 3.1 in (Jiao et al., 2023). Note that

$$\mathbb{E}_{\mathbb{D}_n}[\mathcal{L}_n(\hat{u}) - \mathcal{L}_n(u^0)] \leq \mathbb{E}_{\mathbb{D}_n}[\mathcal{L}_n(u^*) - \mathcal{L}_n(u^0)],$$

which yields that

$$-\mathcal{L}(u^0) \leq 2\mathcal{L}(u^*) - \mathcal{L}(u^0) - \mathbb{E}_{\mathbb{D}_n}[2\mathcal{L}_n(\hat{u})].$$

Then we have

$$\mathbb{E}_{\mathbb{D}_n}[\mathcal{L}(\hat{u}) - \mathcal{L}(u^0)] \leq \mathbb{E}_{\mathbb{D}_n}[\mathcal{L}(\hat{u}) + \mathcal{L}(u^0) - 2\mathcal{L}_n(\hat{u})] + 2[\mathcal{L}(u^*) - \mathcal{L}(u^0)],$$

which completes the proof by using Equation 6.

$\square$

## H.2. Proof of Theorem 6

*Sketch of proof.* This proof is similar to the proof of Theorem 11.4 in (Györfi et al., 2002), where they only consider the squared loss. We first decompose the objective into $\mathcal{D}|\mathcal{S}|$ terms

$$\mathbb{E}_{\mathbb{D}_n}[\mathcal{L}(\hat{u}) + \mathcal{L}(u^0) - 2\mathcal{L}_n(\hat{u})] = \mathbb{E}_{\mathbb{D}_n}\Big[\mathbb{E}_Z[\sum_{d\in[\mathcal{D}],s\in\mathcal{S}}\mathcal{J}(\hat{u},d,s,Z)] - \frac{2}{n}\sum_{i=1}^n\sum_{d\in[\mathcal{D}],s\in\mathcal{S}}\mathcal{J}(\hat{u},d,s,Z_i)\Big]$$

where the function $\mathcal{J}$ is defined in equation 14. To bound the expectation, we focus on the tail probability

$$\mathbb{P}\Big(\exists(u,d,s)\in\mathcal{G}_n\times[\mathcal{D}]\times\mathcal{S}, \mathbb{E}_Z[\mathcal{J}(u,d,s,Z)] - \frac{2}{n}\sum_{i=1}^n\mathcal{J}(u,d,s,Z_i) > \frac{t}{\mathcal{D}|\mathcal{S}|}\Big)$$

Next, we replace the expectation by an empirical mean of a "ghost" sample $\mathbb{D}'_n$ independent of $\mathbb{D}_n$. Additionally, to use concentration inequality, we have to consider the nonnegative empirical mean $\frac{1}{n}\sum_{i=1}^n\big\{\mathcal{J}(u,d,s,Z_i)\big\}^2$ instead of $\frac{1}{n}\sum_{i=1}^n\mathcal{J}(u,d,s,Z_i)$. By symmetrization argument and introducing Rademacher variables $\{\epsilon_i\}_{i=1}^n$, the above probability can be bounded by

$$\mathbb{P}\Big\{\exists(u,d,s)\in\mathcal{G}_n\times[\mathcal{D}]\times\mathcal{S}, \frac{1}{n}\sum_{i=1}^n\epsilon_i\big\{\mathcal{J}(u,d,s,Z_i)\big\}^2 > \Big[\frac{ct}{\mathcal{D}|\mathcal{S}|} + \frac{c}{n}\sum_{i=1}^n\big\{\mathcal{J}(u,d,s,Z_i)\big\}^2\Big]\Big\}$$

$$+ \mathbb{P}\Big(\exists (u, d, s) \in \mathcal{G}_n \times [\mathcal{D}] \times \mathcal{S} : \frac{1}{n} \sum_{i=1}^n \epsilon_i \mathcal{J}(u, d, s, Z_i) > \frac{ct}{\mathcal{D}|\mathcal{S}|} + \frac{c}{n} \sum_{i=1}^n \big\{ \mathcal{J}(u, d, s, Z_i) \big\}^2 \Big).$$

Finally, conditioning on $\mathbb{D}_n$ and introducing a covering, we can apply some concentration inequalities to bound these two terms.

*Proof.* We use the following notation throughout this proof.

$$\mathcal{J}(u, d, s, Z) = \Big(1 - \delta_{X(\mathbf{t})^d}(s)\Big) \Big[ -u_{\mathbf{t}}^d(s, X(\mathbf{t})^d | X(1)^d) \log \frac{u_{\mathbf{t}}^d(s, X(\mathbf{t}))}{u_{\mathbf{t}}^{0,d}(s, X(\mathbf{t}))} + u_{\mathbf{t}}^d(s, X(\mathbf{t})) - u_{\mathbf{t}}^{0,d}(s, X(\mathbf{t})) \Big]. \quad (14)$$

Note that

$$
\begin{aligned}
&\mathbb{E}_{\mathbb{D}_n}[\mathcal{L}(\hat{u}) + \mathcal{L}(u^0) - 2\mathcal{L}_n(\hat{u})] \\
&= \mathbb{E}_{\mathbb{D}_n}[\mathcal{L}(\hat{u}) - \mathcal{L}(u^0) - 2\mathcal{L}_n(\hat{u}) + 2\mathcal{L}(u^0)] \\
&= \mathbb{E}_{\mathbb{D}_n}\Big[\mathbb{E}_Z[\sum_{d \in [\mathcal{D}], s \in \mathcal{S}} \mathcal{J}(\hat{u}, d, s, Z)] - \frac{2}{n} \sum_{i=1}^n \sum_{d \in [\mathcal{D}], s \in \mathcal{S}} \mathcal{J}(\hat{u}, d, s, Z_i)\Big] \\
&\leq a_n + \int_{a_n}^{\infty} \mathbb{P}\Big(\exists (u, d, s) \in \mathcal{G}_n \times [\mathcal{D}] \times \mathcal{S}, \mathbb{E}_Z[\mathcal{J}(u, d, s, Z)] - \frac{2}{n} \sum_{i=1}^n \mathcal{J}(u, d, s, Z_i) > \frac{t}{\mathcal{D}|\mathcal{S}|}\Big) \mathrm{d}t,
\end{aligned}
$$

where $a_n$ is a quantity depending on $n$, which we will choose later. Thus, we can focus on the above tail probability.

**Step 1. Tail probability decomposition.** Following the proof of symmetrization in probability (e.g., Lemma 2.3.7 in (van der Vaart & Wellner, 2023)), consider the following event

$$\mathcal{B}_1 = \{\hat{\mathcal{A}}(t) \text{ is a non-empty set}\},$$

where $t \leq 1$ and

$$
\begin{aligned}
\hat{\mathcal{A}}(t) &= \Big\{(u, d, s) \in \mathcal{G}_n \times [\mathcal{D}] \times \mathcal{S} : \mathbb{E}_Z[\mathcal{J}(u, d, s, Z)] - \frac{2}{n} \sum_{i=1}^n \mathcal{J}(u, d, s, Z_i) > \frac{t}{\mathcal{D}|\mathcal{S}|}\Big\} \\
&= \Big\{(u, d, s) \in \mathcal{G}_n \times [\mathcal{D}] \times \mathcal{S} : \mathbb{E}[\mathcal{J}(u, d, s, Z)] - \frac{1}{n} \sum_{i=1}^n \mathcal{J}(u, d, s, Z_i) \\
&\qquad > \frac{1}{3}\Big(\frac{2t}{\mathcal{D}|\mathcal{S}|} + \mathbb{E}\mathcal{J}(u, d, s, Z) + \frac{1}{n} \sum_{i=1}^n \mathcal{J}(u, d, s, Z_i)\Big)\Big\}.
\end{aligned}
\quad (15)
$$

Let $(\bar{u}, \bar{d}, \bar{s})$ be a (random) function such that $(\bar{u}, \bar{d}, \bar{s}) \in \hat{\mathcal{A}}(t)$ if $\mathcal{B}_1$ holds, and let $(\bar{u}, \bar{d}, \bar{s}) = (1, 1, 1)$ if $\mathcal{B}_1$ does not hold, where $(\bar{u}, \bar{d}, \bar{s})$ depends on $\mathbb{D}_n$. By assumptions, note that $|\mathcal{J}(u, d, s, Z)| \leq (1 + \log \frac{M_\tau}{m}) M_\tau \triangleq K_1$ for any $(u, d, s) \in \mathcal{G}_n \times [\mathcal{D}] \times \mathcal{S}$. Let $\mathbb{D}'_n = \{Z'_i\}_{i=1}^n$ be an independent copy of $\mathbb{D}_n$. By Markov's inequality, we have

$$
\begin{aligned}
&\mathbb{P}\Big\{\mathbb{E}[\mathcal{J}(\bar{u}, \bar{d}, \bar{s}, Z)|\mathbb{D}_n] - \frac{1}{n} \sum_{i=1}^n \mathcal{J}(\bar{u}, \bar{d}, \bar{s}, Z'_i) > \frac{1}{4}\Big[\frac{t}{\mathcal{D}|\mathcal{S}|} + \mathbb{E}[\mathcal{J}(\bar{u}, \bar{d}, \bar{s}, Z)|\mathbb{D}_n]\Big] \Big| \mathbb{D}_n\Big\} \\
&\leq \frac{16\mathbb{E}[(\mathcal{J}(\bar{u}, \bar{d}, \bar{s}, Z))^2|\mathbb{D}_n]}{n(\frac{t}{\mathcal{D}|\mathcal{S}|} + \mathbb{E}[\mathcal{J}(\bar{u}, \bar{d}, \bar{s}, Z)|\mathbb{D}_n])^2} \leq \frac{8K_1 \mathcal{D}|\mathcal{S}|}{nt} \triangleq \eta_1.
\end{aligned}
\quad (16)
$$

Since $\mathcal{B}_1 \in \sigma(\mathbb{D}_n)$, we have

$$(1 - \eta_1)\mathbb{P}\Big(\exists (u, d, s) \in \mathcal{G}_n \times [\mathcal{D}] \times \mathcal{S}, \mathbb{E}_Z[\mathcal{J}(u, d, s, Z)] - \frac{2}{n}\sum_{i=1}^{n}\mathcal{J}(u, d, s, Z_i) > \frac{t}{\mathcal{D}|\mathcal{S}|}\Big)$$

$$\leq \mathbb{E}\Big(\mathbb{1}(\mathcal{B}_1)\mathbb{P}\Big\{\mathbb{E}[\mathcal{J}(\bar{u}, \bar{d}, \bar{s}, Z)|\mathbb{D}_n] - \frac{1}{n}\sum_{i=1}^{n}\mathcal{J}(\bar{u}, \bar{d}, \bar{s}, Z_i') \leq \frac{1}{4}\Big[\frac{t}{\mathcal{D}|\mathcal{S}|} + \mathbb{E}[\mathcal{J}(\bar{u}, \bar{d}, \bar{s}, Z)|\mathbb{D}_n]\Big]\Big|\mathbb{D}_n\Big\}\Big) \qquad (17)$$

$$\leq \mathbb{P}\Big(\exists (u, d, s) \in \mathcal{G}_n \times [\mathcal{D}] \times \mathcal{S} : \frac{1}{n}\sum_{i=1}^{n}\mathcal{J}(u, d, s, Z_i') - \frac{1}{n}\sum_{i=1}^{n}\mathcal{J}(u, d, s, Z_i) > \frac{1}{4}[\frac{t}{\mathcal{D}|\mathcal{S}|} + \mathbb{E}\mathcal{J}(u, d, s, Z)]\Big),$$

where we use equation 16 and the definition of $\hat{A}(t)$ in the second inequality.

Since $\mathcal{J}(u, d, s, Z_i')$ might be negative, to use concentration inequality, we want to focus on the nonnegative quantities $\{\mathcal{J}(u, d, s, Z_i')\}^2$. By a union bound, we can obtain that

$$\mathbb{P}\Big(\exists (u, d, s) \in \mathcal{G}_n \times [\mathcal{D}] \times \mathcal{S} : \frac{1}{n}\sum_{i=1}^{n}\mathcal{J}(u, d, s, Z_i') - \frac{1}{n}\sum_{i=1}^{n}\mathcal{J}(u, d, s, Z_i) > \frac{1}{4}[\frac{t}{\mathcal{D}|\mathcal{S}|} + \mathbb{E}\mathcal{J}(u, d, s, Z)]\Big)$$

$$\leq 2\mathbb{P}\Big(\exists (u, d, s) \in \mathcal{G}_n \times [\mathcal{D}] \times \mathcal{S} : \frac{1}{n}\sum_{i=1}^{n}(\mathcal{J}(u, d, s, Z_i))^2 - \mathbb{E}[\mathcal{J}(u, d, s, Z)]^2$$

$$> \frac{1}{2}\Big[\frac{t}{\mathcal{D}|\mathcal{S}|} + \frac{1}{n}\sum_{i=1}^{n}(\mathcal{J}(u, d, s, Z_i))^2 + \mathbb{E}[\mathcal{J}(u, d, s, Z)]^2\Big]\Big)$$

$$+ \mathbb{P}\Big(\exists (u, d, s) \in \mathcal{G}_n \times [\mathcal{D}] \times \mathcal{S} : \frac{1}{n}\sum_{i=1}^{n}\mathcal{J}(u, d, s, Z_i') - \frac{1}{n}\sum_{i=1}^{n}\mathcal{J}(u, d, s, Z_i) > \frac{1}{4}[\frac{t}{\mathcal{D}|\mathcal{S}|} + \mathbb{E}\mathcal{J}(u, d, s, Z)],$$

$$\frac{1}{n}\sum_{i=1}^{n}(\mathcal{J}(u, d, s, Z_i'))^2 - \mathbb{E}[\mathcal{J}(u, d, s, Z)]^2 \leq \frac{1}{2}\Big[\frac{t}{\mathcal{D}|\mathcal{S}|} + \frac{1}{n}\sum_{i=1}^{n}(\mathcal{J}(u, d, s, Z_i'))^2 + \mathbb{E}[\mathcal{J}(u, d, s, Z)]^2\Big],$$

$$\frac{1}{n}\sum_{i=1}^{n}(\mathcal{J}(u, d, s, Z_i))^2 - \mathbb{E}[\mathcal{J}(u, d, s, Z)]^2 \leq \frac{1}{2}\Big[\frac{t}{\mathcal{D}|\mathcal{S}|} + \frac{1}{n}\sum_{i=1}^{n}(\mathcal{J}(u, d, s, Z_i))^2 + \mathbb{E}[\mathcal{J}(u, d, s, Z)]^2\Big]\Big)$$

$$\stackrel{\triangle}{=} 2\mathbb{P}(\mathcal{B}_2) + \mathbb{P}(\mathcal{B}_3).$$

**Step 2. Bounding $\mathbb{P}(\mathcal{B}_2)$.** Let $(\bar{u}, \bar{d}, \bar{s}) \in \mathcal{G}_n \times [\mathcal{D}]] \times \mathcal{S}$ be a function such that

$$\frac{1}{n}\sum_{i=1}^{n}(\mathcal{J}(\bar{u}, \bar{d}, \bar{s}, Z_i))^2 - \mathbb{E}[\mathcal{J}(\bar{u}, \bar{d}, \bar{s}, Z)]^2 > \frac{1}{2}\Big[\frac{t}{\mathcal{D}|\mathcal{S}|} + \frac{1}{n}\sum_{i=1}^{n}(\mathcal{J}(\bar{u}, \bar{d}, \bar{s}, Z_i))^2 + \mathbb{E}[\mathcal{J}(\bar{u}, \bar{d}, \bar{s}, Z)]^2\Big],$$

if $\mathcal{B}_2$ holds; and $(\bar{u}, \bar{d}, \bar{s}) = (1, 1, 1)$ otherwise, which depends on $\mathbb{D}_n$ Conditioning on $\mathcal{B}_2$ and $\mathbb{D}_n$, by Markov's inequality, we have

$$\mathbb{P}\Big\{\frac{1}{n}\sum_{i=1}^{n}(\mathcal{J}(\bar{u}, \bar{d}, \bar{s}, Z_i'))^2 - \mathbb{E}[(\mathcal{J}(\bar{u}, \bar{d}, \bar{s}, Z))^2|\mathbb{D}_n] > \frac{1}{8}\Big[\frac{t}{\mathcal{D}|\mathcal{S}|} + \frac{1}{n}\sum_{i=1}^{n}(\mathcal{J}(\bar{u}, \bar{d}, \bar{s}, Z_i'))^2 + \mathbb{E}[\mathcal{J}^2(\bar{u}, \bar{d}, \bar{s}, Z)|\mathbb{D}]\Big]\Big|\mathbb{D}_n\Big\}$$

$$= \mathbb{P}\Big\{\frac{7}{n}\sum_{i=1}^{n}(\mathcal{J}(\bar{u}, \bar{d}, \bar{s}, Z_i'))^2 - 7\mathbb{E}[(\mathcal{J}(\bar{u}, \bar{d}, \bar{s}, Z))^2|\mathbb{D}_n] > \frac{t}{\mathcal{D}|\mathcal{S}|} + 2\mathbb{E}[(\mathcal{J}(\bar{u}, \bar{d}, \bar{s}, Z))^2|\mathbb{D}_n]\Big|\mathbb{D}_n\Big\}$$

$$\leq \frac{49\mathbb{E}[\mathcal{J}^4(\bar{u}, \bar{d}, \bar{s}, Z)|\mathbb{D}_n]}{n(\frac{t}{\mathcal{D}|\mathcal{S}|} + 2\mathbb{E}[(\mathcal{J}(\bar{u}, \bar{d}, \bar{s}, Z))^2|\mathbb{D}_n])^2} \leq \frac{49K_1^2|\mathcal{S}|\mathcal{D}}{4nt} \stackrel{\triangle}{=} \eta_2.$$

$$(18)$$

Then, by using equation 18 and a symmetrization argument, we have

$$(1 - \eta_2)\mathbb{P}\Big(\mathcal{B}_2\Big)$$

$$\leq \mathbb{E}\Big(\mathbb{1}(\mathcal{B}_2)\mathbb{P}\Big\{\frac{1}{n}\sum_{i=1}^{n}\Big\{\mathcal{J}(\bar{u},\bar{d},\bar{s},Z_i')\Big\}^2 - \mathbb{E}[(\mathcal{J}(\bar{u},\bar{d},\bar{s},Z))^2|\mathbb{D}_n]$$

$$\leq \frac{1}{8}\Big[\frac{t}{\mathcal{D}|\mathcal{S}|} + \frac{1}{n}\sum_{i=1}^{n}\Big\{\mathcal{J}(\bar{u},\bar{d},\bar{s},Z_i')\Big\}^2 + \mathbb{E}[(\mathcal{J}(\bar{u},\bar{d},\bar{s},Z))^2|\mathbb{D}_n]\Big]\Big|\mathbb{D}_n\Big\}\Big)$$

$$\leq \mathbb{P}\Big(\exists (u,d,s)\in \mathcal{G}_n\times[\mathcal{D}]\times\mathcal{S}, \frac{4}{n}\sum_{i=1}^{n}\Big\{\mathcal{J}(u,d,s,Z_i)\Big\}^2 - \frac{7}{n}\sum_{i=1}^{n}\Big\{\mathcal{J}(u,d,s,Z_i')\Big\}^2 > 3\Big[\frac{t}{\mathcal{D}|\mathcal{S}|} + \mathbb{E}[\mathcal{J}(u,d,s,Z)]^2\Big]\Big)$$

$$= \mathbb{P}\Big(\exists (u,d,s)\in \mathcal{G}_n\times[\mathcal{D}]\times\mathcal{S}, \frac{1}{n}\sum_{i=1}^{n}\Big\{\mathcal{J}(u,d,s,Z_i')\Big\}^2 - \frac{1}{n}\sum_{i=1}^{n}\Big\{\mathcal{J}(u,d,s,Z_i)\Big\}^2$$

$$> \frac{3}{11}\Big[\frac{2t}{\mathcal{D}|\mathcal{S}|} + 2\mathbb{E}[\mathcal{J}(u,d,s,Z)]^2 + \frac{1}{n}\sum_{i=1}^{n}\Big\{\mathcal{J}(u,d,s,Z_i')\Big\}^2 + \frac{1}{n}\sum_{i=1}^{n}\Big\{\mathcal{J}(u,d,s,Z_i)\Big\}^2\Big]\Big)$$

$$\leq \mathbb{P}\Big(\exists (u,d,s)\in \mathcal{G}_n\times[\mathcal{D}]\times\mathcal{S}, \frac{1}{n}\sum_{i=1}^{n}\Big\{\mathcal{J}(u,d,s,Z_i')\Big\}^2 - \frac{1}{n}\sum_{i=1}^{n}\Big\{\mathcal{J}(u,d,s,Z_i)\Big\}^2$$

$$> \frac{3}{11}\Big[\frac{2t}{\mathcal{D}|\mathcal{S}|} + \frac{1}{n}\sum_{i=1}^{n}\Big\{\mathcal{J}(u,d,s,Z_i')\Big\}^2 + \frac{1}{n}\sum_{i=1}^{n}\Big\{\mathcal{J}(u,d,s,Z_i)\Big\}^2\Big]\Big)$$

$$= \mathbb{P}\Big(\exists (u,d,s)\in \mathcal{G}_n\times[\mathcal{D}]\times\mathcal{S}, \frac{1}{n}\sum_{i=1}^{n}\epsilon_i\Big\{\mathcal{J}(u,d,s,Z_i')\Big\}^2 - \frac{1}{n}\sum_{i=1}^{n}\epsilon_i\Big\{\mathcal{J}(u,d,s,Z_i)\Big\}^2$$

$$> \frac{3}{11}\Big[\frac{2t}{\mathcal{D}|\mathcal{S}|} + \frac{1}{n}\sum_{i=1}^{n}\Big\{\mathcal{J}(u,d,s,Z_i')\Big\}^2 + \frac{1}{n}\sum_{i=1}^{n}\Big\{\mathcal{J}(u,d,s,Z_i)\Big\}^2\Big]\Big)$$

$$\leq 2\mathbb{P}\Big\{\exists (u,d,s)\in \mathcal{G}_n\times[\mathcal{D}]\times\mathcal{S}, \frac{1}{n}\sum_{i=1}^{n}\epsilon_i\Big\{\mathcal{J}(u,d,s,Z_i)\Big\}^2 > \frac{3}{11}\Big[\frac{t}{\mathcal{D}|\mathcal{S}|} + \frac{1}{n}\sum_{i=1}^{n}\Big\{\mathcal{J}(u,d,s,Z_i)\Big\}^2\Big]\Big\},$$

where $\{\epsilon_i\}_{i=1}^{n}$ is a sequence of i.i.d. Rademacher random variables, independent of $\mathbb{D}_n$.

Given $\mathbb{D}_n$, we consider the following sequence with length $n$:

$$\mathbf{s}_n = \Big\{(\mathbf{t}_i, X_i(\mathbf{t}_i))\Big\}_{i\in[n]},$$

Let $\mathcal{H}_\delta^{d,s}(\mathbb{D}_n)$ be a $L^\infty$ $\delta$-covering of $\mathcal{G}_n'|_{\mathbf{s}_n}$ with minimal size. That is to say, given $\mathbb{D}_n$, $d\in[\mathcal{D}]$ and $s\in\mathcal{S}$, for any $u_\cdot^d(s,\cdot)\in\mathcal{G}_n'$, there exists $\tilde{u}_\cdot^d(s,\cdot)\in\mathcal{H}_\delta^{d,s}(\mathbb{D}_n)$ such that for any $i\in[n]$

$$|u_{\mathbf{t}_i}^d(s, X_i(\mathbf{t}_i)) - \tilde{u}_{\mathbf{t}_i}^d(s, X_i(\mathbf{t}_i))| \leq \delta,$$

which implies that

$$|\mathcal{J}(u,d,s,Z_i) - \mathcal{J}(\tilde{u},d,s,Z_i)| \leq (\frac{M_\tau}{m} + 1)\delta \overset{\triangle}{=} K_2\delta, \tag{19}$$

and

$$|\mathcal{J}^2(u,d,s,Z_i) - \mathcal{J}^2(\tilde{u},d,s,Z_i)| \leq 2K_1K_2\delta \overset{\triangle}{=} \eta_3\delta. \tag{20}$$

Define

$$\mathcal{H}_\delta(\mathbb{D}_n) = \Big\{\{u_\cdot^d(s,\cdot)\}_{d\in\mathcal{D},s\in\mathcal{S}} : \mathcal{S}^\mathcal{D}\times[0,1]\to\mathbb{R}^{\mathcal{D}\times|\mathcal{S}|} : |u_t^d(z^d,x)|\geq m, u_\cdot^d(s,\cdot)\in\mathcal{H}_\delta^{d,s}(\mathbb{D}_n)\Big\}.$$

Thus, taking $\delta_1 = \frac{t}{7\eta_3\mathcal{D}|\mathcal{S}|}$, by union bound, we have

$$\mathbb{P}\Big(\mathcal{B}_2\Big)$$

$$\leq \frac{2}{1-\eta_2}\mathbb{P}\Big(\exists(u,d,s)\in\mathcal{G}_n\times[\mathcal{D}]\times\mathcal{S},\frac{1}{n}\sum_{i=1}^{n}\epsilon_i\Big\{\mathcal{J}(u,d,s,Z_i)\Big\}^2 > \frac{3}{11}\Big[\frac{t}{\mathcal{D}|\mathcal{S}|}+\frac{1}{n}\sum_{i=1}^{n}\Big\{\mathcal{J}(u,d,s,Z_i)\Big\}^2\Big]\Big)$$

$$\leq \frac{2}{1-\eta_2}\mathbb{P}\Big(\exists(u,d,s)\in\mathcal{H}_{\delta_1}(\mathbb{D}_n)\times[\mathcal{D}]\times\mathcal{S},\eta_3\delta_1+\frac{1}{n}\sum_{i=1}^{n}\epsilon_i\Big\{\mathcal{J}(u,d,s,Z_i)\Big\}^2$$

$$> \frac{3}{11}\Big[\frac{t}{\mathcal{D}|\mathcal{S}|}+\frac{1}{n}\sum_{i=1}^{n}\Big\{\mathcal{J}(u,d,s,Z_i)\Big\}^2-\eta_3\delta_1\Big]\Big)$$

$$= \frac{2}{1-\eta_2}\mathbb{P}\Big(\exists(u,d,s)\in\mathcal{H}_{\delta_1}(\mathbb{D}_n)\times[\mathcal{D}]\times\mathcal{S},\frac{1}{n}\sum_{i=1}^{n}\epsilon_i\Big\{\mathcal{J}(u,d,s,Z_i)\Big\}^2$$

$$> \frac{3t}{11\mathcal{D}|\mathcal{S}|}+\frac{3}{11n}\sum_{i=1}^{n}\Big\{\mathcal{J}(u,d,s,Z_i)\Big\}^2-\frac{14}{11}\eta_3\delta_1\Big)$$

$$= \frac{2}{1-\eta_2}\mathbb{P}\Big(\exists(u,d,s)\in\mathcal{H}_{\delta_1}(\mathbb{D}_n)\times[\mathcal{D}]\times\mathcal{S},\frac{1}{n}\sum_{i=1}^{n}\epsilon_i\Big\{\mathcal{J}(u,d,s,Z_i)\Big\}^2 > \frac{t}{11\mathcal{D}|\mathcal{S}|}+\frac{3}{11n}\sum_{i=1}^{n}\Big\{\mathcal{J}(u,d,s,Z_i)\Big\}^2\Big).$$

Conditioning on $\mathbb{D}_n$, by Hoeffding's inequality, we have

$$\mathbb{E}\Big\{\mathbb{P}\Big(\exists(u,d,s)\in\mathcal{H}_{\delta_1}(\mathbb{D}_n)\times[\mathcal{D}]\times\mathcal{S},\frac{1}{n}\sum_{i=1}^{n}\epsilon_i\Big\{\mathcal{J}(u,d,s,Z_i)\Big\}^2 > \frac{t}{11\mathcal{D}|\mathcal{S}|}+\frac{3}{11n}\sum_{i=1}^{n}\Big\{\mathcal{J}(u,d,s,Z_i)\Big\}^2\Big|\mathbb{D}_n\Big)\Big\}$$

$$\leq \mathbb{E}\Big\{2|\mathcal{S}|\mathcal{D}\mathcal{N}_n(\delta_1,\mathcal{G}_n'|_{\mathbf{s}_n},L^\infty)]\times\max_{u\in\mathcal{H}_{\delta_1}(\mathbb{D}_n)}\exp\Big(-\frac{cn\Big(\frac{t}{11\mathcal{D}|\mathcal{S}|}+\frac{3}{11n}\sum_{i=1}^{n}\Big\{\mathcal{J}(u,d,s,Z_i)\Big\}^2\Big)^2}{\frac{1}{n}\sum_{i=1}^{n}\Big\{\mathcal{J}(u,d,s,Z_i)\Big\}^4}\Big)\Big\}$$

$$\leq 2\mathcal{D}|\mathcal{S}|\mathcal{N}_n(\frac{t}{7\eta_3\mathcal{D}|\mathcal{S}|},\mathcal{G}_n',L^\infty)\exp(-\frac{cnt}{K_1^2|\mathcal{S}|\mathcal{D}}).$$

Consequently, we have

$$\mathbb{P}(\mathcal{B}_2) \leq \Big(\frac{4\mathcal{D}|\mathcal{S}|}{1-\eta_2}\Big)\mathcal{N}_n(\frac{t}{7\eta_3\mathcal{D}},\mathcal{G}_n',L^\infty)\exp(-\frac{cnt}{K_1^2|\mathcal{S}|\mathcal{D}}). \tag{21}$$

**Step 3. Bounding $\mathbb{P}(\mathcal{B}_3)$.** For $F(x)=x\log x$, if $a,b\in[c,C]$, then we have $D_F(a||b)\geq\frac{1}{2C}(a-b)^2$. Thus, for any $(u,d)\in\mathcal{G}_n\times[D]$, by equation 20 and QM-AM inequality, we have

$$\mathbb{E}\Big[\mathcal{J}(u,d,s,Z)\Big] = \mathbb{E}\Big[(1-\delta_{X(\mathbf{t})^d}(s))D_F(u_{\mathbf{t}}^{0,d}(s,X(\mathbf{t}))||u_{\mathbf{t}}^d(s,X(\mathbf{t})))\Big]$$

$$\geq \frac{1}{M_\tau}\mathbb{E}\Big[(1-\delta_{X(\mathbf{t})^d}(s))(u_{\mathbf{t}}^{0,d}(s,X(\mathbf{t}))-u_{\mathbf{t}}^d(s,X(\mathbf{t})))^2\Big]$$

$$\geq \frac{1}{M_\tau K_2^2}\mathbb{E}\Big[(1-\delta_{X(\mathbf{t})^d}(s))D_F^2(u_{\mathbf{t}}^{0,d}(s,X(\mathbf{t}))||u_{\mathbf{t}}^d(s,X(\mathbf{t})))\Big] \tag{22}$$

$$\geq \frac{1}{M_\tau K_2^2}\mathbb{E}\Big[\mathcal{J}^2(u,d,s,Z)\Big]$$

$$\geq C\alpha\mathbb{E}\Big[\mathcal{J}^2(u,d,s,Z)\Big]^2,$$

where $\alpha=(M_\tau K_2^2)^{-1}\wedge(1/2)$. Then, by the definition of $\mathcal{B}_3$, equation 22 and introducing random signs, we have

$$\mathbb{P}(\mathcal{B}_3) \leq \mathbb{P}\Big(\exists(u,d,s)\in\mathcal{G}_n\times[\mathcal{D}]\times\mathcal{S}:\frac{1}{n}\sum_{i=1}^{n}\mathcal{J}(u,d,s,Z_i')-\frac{1}{n}\sum_{i=1}^{n}\mathcal{J}(u,d,s,Z_i)$$

$$> \Big(\frac{1}{2}-\frac{C\alpha}{12}\Big)\frac{t}{\mathcal{D}|\mathcal{S}|}+\frac{C\alpha}{24}\Big(\frac{1}{n}\sum_{i=1}^{n}\Big\{\mathcal{J}(u,d,s,Z_i')\Big\}^2+\frac{1}{n}\sum_{i=1}^{n}\Big\{\mathcal{J}(u,d,s,Z_i)\Big\}^2\Big)\Big)$$

$$\leq 2\mathbb{P}\Big(\exists (u,d,s) \in \mathcal{G}_n \times [\mathcal{D}] \times \mathcal{S} : \frac{1}{n}\sum_{i=1}^{n}\epsilon_i \mathcal{J}(u,d,s,Z_i) > \Big(\frac{1}{4} - \frac{C\alpha}{24}\Big)\frac{t}{\mathcal{D}|\mathcal{S}|} + \frac{C\alpha}{24n}\sum_{i=1}^{n}\Big\{\mathcal{J}(u,d,s,Z_i)\Big\}^2\Big),$$

where $\{\epsilon_i\}_{i=1}^n$ is a sequence of i.i.d. Rademacher random variables, independent of $\mathbb{D}_n$. Note that, given $\mathbb{D}_n$, for any $u \in \mathcal{G}_n$, there exists $\tilde{u} \in \mathcal{H}_\delta(\mathbb{D}_n)$ such that for any $i \in [n]$,

$$\Big|\mathcal{J}(u,d,s,Z_i) - \mathcal{J}(\tilde{u},d,s,Z_i)\Big| \leq K_2\delta \overset{\triangle}{=} \eta_4\delta.$$

Thus, if $\delta_2 = \frac{t}{(16\eta_4 + 2\eta_3)\mathcal{D}|\mathcal{S}|}$, conditioning on $\mathbb{D}_n$, by union bound and Bernstein's inequality (e.g., Lemma A.2 of Györfi et al., 2002), we have

$$2\mathbb{P}\Big(\exists (u,d,s) \in \mathcal{G}_n \times [\mathcal{D}] \times \mathcal{S} : \frac{1}{n}\sum_{i=1}^{n}\epsilon_i \mathcal{J}(u,d,s,Z_i) > \Big(\frac{1}{4} - \frac{C\alpha}{24}\Big)\frac{t}{\mathcal{D}|\mathcal{S}|} + \frac{C\alpha}{24n}\sum_{i=1}^{n}\Big\{\mathcal{J}(u,d,s,Z_i)\Big\}^2\Big)$$

$$= 2\mathbb{E}\Big\{\mathbb{P}\Big(\exists (u,d,s) \in \mathcal{G}_n \times [\mathcal{D}] \times \mathcal{S} : \frac{1}{n}\sum_{i=1}^{n}\epsilon_i \mathcal{J}(u,d,s,Z_i) > \Big(\frac{1}{4} - \frac{C\alpha}{24}\Big)\frac{t}{\mathcal{D}} + \frac{C\alpha}{24n}\sum_{i=1}^{n}\Big\{\mathcal{J}(u,d,s,Z_i)\Big\}^2\Big|\mathbb{D}_n\Big)\Big\}$$

$$\leq 2\mathbb{E}\Big\{\mathbb{P}\Big(\exists (u,d,s) \in \mathcal{H}_\delta(\mathbb{D}_n) \times [\mathcal{D}] \times \mathcal{S} : \eta_4\delta_2 + \frac{1}{n}\sum_{i=1}^{n}\epsilon_i \mathcal{J}(u,d,s,Z_i)$$

$$> \Big(\frac{1}{4} - \frac{C\alpha}{24}\Big)\frac{t}{\mathcal{D}|\mathcal{S}|} + \frac{C\alpha}{24n}\sum_{i=1}^{n}\Big\{\mathcal{J}(u,d,s,Z_i)\Big\}^2 - \frac{C\alpha\eta_3\delta_2}{24}\Big|\mathbb{D}_n\Big)\Big\}$$

$$\leq 2\mathbb{E}\Big\{\mathcal{D}|\mathcal{S}|\mathcal{N}_n(\delta_2,\mathcal{G}'_n|_{\mathbf{s}_n},L^\infty)\max_{u \in \mathcal{H}_\delta(\mathbb{D}_n)}\mathbb{P}\Big(\frac{1}{n}\sum_{i=1}^{n}\epsilon_i \mathcal{J}(u,d,s,Z_i)$$

$$> \Big(\frac{3}{16} - \frac{C\alpha}{24}\Big)\frac{t}{\mathcal{D}|\mathcal{S}|} + \frac{C\alpha}{24n}\sum_{i=1}^{n}\Big\{\mathcal{J}(u,d,s,Z_i)\Big\}^2\Big|\mathbb{D}_n\Big)\Big\}$$

$$\leq 4\mathbb{E}\Big\{\mathcal{D}|\mathcal{S}|\mathcal{N}_n(\delta_2,\mathcal{G}'_n|_{\mathbf{s}_n},L^\infty)\max_{u \in \mathcal{H}_\delta(\mathbb{D}_n)}\exp\Big(-\frac{cn\Big[\Big(\frac{3}{16} - \frac{C\alpha}{24}\Big)\frac{t}{\mathcal{D}|\mathcal{S}|} + \frac{C\alpha}{24n}\sum_{i=1}^{n}\Big\{\mathcal{J}(u,d,s,Z_i)\Big\}^2\Big]^2}{\frac{2}{n}\sum_{i=1}^{n}\Big\{\mathcal{J}(u,d,s,Z_i)\Big\}^2 + \frac{2t}{\mathcal{D}|\mathcal{S}|}}\Big)\Big\}$$

$$\leq 4\mathcal{D}|\mathcal{S}|\mathcal{N}_n\Big(\frac{t}{(16\eta_4 + 2\eta_3)\mathcal{D}|\mathcal{S}|},\mathcal{G}'_n,L^\infty\Big)\exp\Big(-\frac{c\alpha nt}{\mathcal{D}|\mathcal{S}|}\Big),$$

where the last inequality we use the inequality

$$\frac{(a+u)^2}{a+bu} \geq \frac{4a}{b^2}[(b-1)\vee 0] \text{ for any } a,b,u > 0.$$

Consequently, we have

$$\mathbb{P}(\mathcal{B}_3) \leq 4\mathcal{D}|\mathcal{S}|\mathcal{N}_{|\mathcal{S}|n}\Big(\frac{t}{(16\eta_4 + 2\eta_3)\mathcal{D}|\mathcal{S}|},\mathcal{G}'_n,L^\infty\Big)\exp\Big(-\frac{c\alpha nt}{\mathcal{D}}\Big). \tag{23}$$

**Step 4. Deriving the final bound.** Combining Equation 21 and Equation 23 derived in previous parts, if $t \geq CK_1^2|\mathcal{S}|\mathcal{D}/n$, we have

$$\mathbb{P}\Big(\exists (u,d,s) \in \mathcal{G}_n \times [\mathcal{D}] \times \mathcal{S}, \mathbb{E}_Z[\mathcal{J}(u,d,s,Z)] - \frac{2}{n}\sum_{i=1}^{n}\mathcal{J}(u,d,s,Z_i) > \frac{t}{\mathcal{D}|\mathcal{S}|}\Big)$$

$$\leq \frac{1}{1-\eta_1}(2\mathbb{P}(\mathcal{B}_2) + \mathbb{P}(\mathcal{B}_3))$$

$$\leq C\mathcal{D}\mathcal{N}_n\Big(\frac{t}{CK_1^2|\mathcal{S}|\mathcal{D}},\mathcal{G}'_n,L^\infty\Big)\exp\Big(-\frac{cnt}{K_1^2|\mathcal{S}|^2\mathcal{D}}\Big).$$

Thus, by taking $a_n = \frac{CK_1^2|\mathcal{S}|\mathcal{D}[\log \mathcal{N}_n(1/(2n),\mathcal{G}'_n,L^\infty)+\log(\mathcal{D}|\mathcal{S}|)]}{n}$, we can obtain that

$$
\begin{aligned}
&\mathbb{E}_{\mathbb{D}_n}[\mathcal{L}(\hat{u}) + \mathcal{L}(u^0) - 2\mathcal{L}_n(\hat{u})] \\
&\leq a_n + \int_{a_n}^\infty \mathbb{P}\Big(\exists (u,d,s) \in \mathcal{G}_n \times [\mathcal{D}] \times \mathcal{S}, \mathbb{E}_Z[\mathcal{J}(u,d,s,Z)] - \frac{2}{n}\sum_{i=1}^n \mathcal{J}(u,d,s,Z_i) > \frac{t}{\mathcal{D}|\mathcal{S}|}\Big) \mathrm{d}t \\
&\leq a_n + \int_{a_n}^\infty \exp\Big(-\frac{cnt}{K_1^2|\mathcal{S}|\mathcal{D}} + \log \mathcal{N}_n(1/(2n),\mathcal{G}'_n,L^\infty) + \log(\mathcal{D}|\mathcal{S}|)\Big) \mathrm{d}t \\
&\leq a_n + \frac{CK_1^2|\mathcal{S}|\mathcal{D}}{n} \\
&\leq \frac{CK_1^2|\mathcal{S}|\mathcal{D}[\log \mathcal{N}_n(1/(2n),\mathcal{G}'_n,L^\infty) + \log(\mathcal{D}|\mathcal{S}|)]}{n},
\end{aligned}
$$

which completes the proof by noting that $K_1 = (1 + \log \frac{M_\tau}{m})M_\tau$.

$\square$

### H.3. Proof of Theorem 7

*Proof.* We first establish the approximation error under the boundedness condition (Assumption 2), then we use the hyperparameters in the proof of Theorem 5 to derive the distribution error bound.

**Step 1. Bounding approximation error.** By Assumption 2, we can derive a new bound for the LHS of equation 12

$$
\begin{aligned}
&\inf_{u\in\mathcal{G}_n} \mathbb{E}\Big[\sum_{z\neq X(\mathbf{t})} D_F\Big(u_{\mathbf{t}}^0(z,X(\mathbf{t}))\|u_{\mathbf{t}}(z,X(\mathbf{t}))\Big)\Big] \\
&\leq m^{-1}\mathbb{E}\Big\{\sum_{d\in[\mathcal{D}],s\in\mathcal{S}} \Big|f^{***}(d,s,\mathbf{t},X(\mathbf{t})) - f^{0,d,s}(\mathbf{t},X(\mathbf{t}))\Big|^2\Big\} \\
&\leq C\mathcal{D}|\mathcal{S}|m^{-1}|\mathcal{S}|^{2\beta}B_0^2(\lfloor\beta\rfloor+1)^4(\mathcal{D}+1)^{2\lfloor\beta\rfloor+(\beta\vee 1)}(S_1 S_2)^{-4\beta/(\mathcal{D}+1)},
\end{aligned}
\tag{24}
$$

where $\mathcal{G}_n$ is the class of matrix-valued ReLU networks with range $[m_n, M_\tau]^{\mathcal{D}\times|\mathcal{S}|}$, depth $\mathrm{D}^* = 21(\lfloor\beta\rfloor+1)^2 S_1\lceil\log_2(8S_1)\rceil + 3$, width $\mathrm{W}^* = 38(\lfloor\beta\rfloor+1)^2(\mathcal{D}+1)^{\lfloor\beta\rfloor+1}S_2\lceil\log_2(8S_2)\rceil$, $S_1, S_2 \in \mathbb{N}^+$.

**Step 2. Bounding estimation error.** Given the vocabulary size $|\mathcal{S}|$, by Theorem 6, Lemma 6 and Lemma 7, the stochastic error is bounded by

$$
\mathbb{E}_{\mathbb{D}_n}[\mathcal{L}(\hat{u}) + \mathcal{L}(u^0) - 2\mathcal{L}_n(\hat{u})] \leq C(\log n\tau^{-1})^2|\mathcal{S}|\mathcal{D}\frac{\mathrm{S}^*\mathrm{D}^*\log\mathrm{S}^*}{n\tau^2},
$$

where $\mathrm{S}^*$ is the number of parameters of the ReLU networks in $\mathcal{G}'_n$. Note that for a ReLU network with depth $\mathrm{D}^*$ and width $\mathrm{W}^*$ and input dimension $\mathcal{D}+1$, we have (assume that $\mathcal{D} \lesssim \mathrm{W}^*\mathrm{D}^*$)

$$
\mathrm{S}^* \leq \underbrace{\mathrm{W}^*(\mathcal{D}+1) + \mathrm{W}^*}_{\text{input layer}} + \underbrace{((\mathrm{W}^*)^2 + \mathrm{W}^*)(\mathrm{D}^* - 1)}_{\text{hidden layer}} + \underbrace{\mathrm{W}^* + 1}_{\text{output layer}} = O((\mathrm{W}^*)^2\mathrm{D}^*).
$$

Therefore, choosing $S_1 = C(n\tau^2)^{\frac{\mathcal{D}+1}{(2\mathcal{D}+4\beta+2)}}$ and $S_2 = C$, we have

$$
\mathrm{W}^* = C(\mathcal{D}+1)^{\lfloor\beta\rfloor+1}; \quad \mathrm{D}^* = C(n\tau^2)^{\frac{\mathcal{D}+1}{(2\mathcal{D}+4\beta+2)}}\log(n\tau^2); \quad \mathrm{S}^* = C\mathcal{D}^{2\lfloor\beta\rfloor+2}(n\tau^2)^{\frac{\mathcal{D}+1}{(2\mathcal{D}+4\beta+2)}}\log(n\tau^2),
$$

yielding that

$$
\mathbb{E}_{\mathbb{D}_n}[\mathcal{L}(\hat{u}) + \mathcal{L}(u^0) - 2\mathcal{L}_n(\hat{u})] \leq C|\mathcal{S}|\mathcal{D}^{2\lfloor\beta\rfloor+3}(n\tau^2)^{-\frac{2\beta}{(2\beta+\mathcal{D}+1)}}\log(n\tau^2)\log(n\tau^{-1}),
$$

if $n \geq \mathcal{D}$.

By equation 24, the approximation error is

$$\inf_{u \in \mathcal{G}_n} \mathbb{E}\Big[D_F\Big(\sum_{z \neq X(\mathbf{t})} u^0(\mathbf{t}, z, X(\mathbf{t})) \| \sum_{z \neq X(\mathbf{t})} u(\mathbf{t}, z, X(\mathbf{t}))\Big)\Big] \leq C|\mathcal{S}|^{2\beta+1}\mathcal{D}^{2\lfloor\beta\rfloor+3}\tau^{-2}(n\tau^2)^{-\frac{2\beta}{(2\beta+\mathcal{D}+1)}}$$

Then the summation of the approximation error and the stochastic error has the convergence rate (up to some logarithmic multiplier)

$$\mathbb{E}_{\mathbb{D}_n}[D_{KL}(p_{1-\tau}\|\hat{p}_{1-\tau})] \leq C|\mathcal{S}|^{2\beta+1}\mathcal{D}^{2\lfloor\beta\rfloor+3}\tau^{-2}(n\tau^2)^{-\frac{2\beta}{(2\beta+\mathcal{D}+1)}}. \tag{25}$$

**Step 3. Deriving final results.** By triangle inequality, Pinsker's inequality and Theorems 2 and 3, for sufficiently large $n\tau^2$, we have

$$\begin{aligned}
\mathbb{E}_{\mathbb{D}_n}\Big[\text{TV}(p_1, \hat{p}_{1-\tau})\Big] &\leq \mathbb{E}_{\mathbb{D}_n}\Big[\text{TV}(p_{1-\tau}, \hat{p}_{1-\tau})\Big] + \text{TV}(p_1, p_{1-\tau}) \\
&\leq \mathbb{E}_{\mathbb{D}_n}[D_{KL}(p_{1-\tau}\|\hat{p}_{1-\tau})] + \text{TV}(p_1, p_{1-\tau}) \\
&\leq C|\mathcal{S}|^{\beta+1}\mathcal{D}^{\lfloor\beta\rfloor+\frac{3}{2}}\tau^{-1}(n\tau^2)^{-\frac{\beta}{(2\beta+\mathcal{D}+1)}} + C\tau\mathcal{D}
\end{aligned}$$

which completes the proof.

$$\square$$

# I. Implementation Details and Additional Experiments

## I.1. Implementation Details

**Data.** We consider the data distribution with a blockwise AR(1) structure; that is, we first sample dimension $d = 1$ from $X(1)^1 \sim \mathcal{U}(\mathcal{S})$ and then we sample dimension $d = 2, 3$ from

$$X(1)^d | X(1)^{d-1} \sim \begin{cases} 0.9\,\mathcal{U}(X(1)^{d-1} + \{-2, -1, \ldots, 2\}) + 0.1\,\mathcal{U}(\mathcal{S}), & \text{if } X(1)^{d-1} \in [3, |\mathcal{S}| - 2] \\ \mathcal{U}(\mathcal{S}), & \text{otherwise} \end{cases},$$

and finally we sample $(X(1)^{3j-2}, X(1)^{3j-1}, X(1)^{3j})$ from distribution same as $(X(1)^1, X(1)^2, X(1)^3)$ for $j = 2, \ldots, \mathcal{D}/3$ (if $\mathcal{D} > 3$).

**Experimental Setup.** We consider different sample size, dimension and early stopping parameter in our experiments.

- sample size: $n \in \{2500, 5000, 7500, 10000, 12500\}$;
- dimension: $\mathcal{D} \in \{3, 6, 9, 12, 15\}$;
- early stopping parameter: $\tau \in \{0.001, 0.003, 0.005, 0.007, 0.01, 0.03, 0.05, 0.07, 0.1\}$.

**Training and Evaluation.** In our experiments, we consider the linear time scheduler $\kappa_t = t$. Note that $u_t^d(z^d, x) = \frac{1}{1-t}(p_{1|t}^d(z^d|x) - \delta_{x^d}(z^d))$ by equation 3 and equation 4. Thus, we can parameterize the posterior $p_{1|t}^{d,\theta}(z^d|x)$. By equation 4, the estimated transition rate is $\hat{u}_t(z, x) = \sum_{d=1}^{\mathcal{D}} \frac{1}{1-t}\delta_{x^{\backslash d}}(z^{\backslash d})(p_{1|t}^{d,\theta}(z^d|x) - \delta_{x^d}(z^d))$, where the parameters can be obtained by minimizing the following empirical risk (equivalent to the empirical version of ELBO derived in Equation 37 of (Shaul et al., 2025)) based on data $\mathbb{D}_n$, which is equivalent to equation 5 (up to a constant not depending on $\theta$):

$$-\frac{1}{n}\sum_{i=1}^{n}\sum_{d=1}^{\mathcal{D}}\frac{1}{1-\mathbf{t}_i}\Big\{\Big(1 - \delta_{X_i(1)^d}(X_i(\mathbf{t}_i)^d)\Big)\log p_{1|t}^{d,\theta}(X_i(1)^d|X_i(\mathbf{t}_i)) - \delta_{X_i(1)^d}(X_i(\mathbf{t}_i)^d) + p_{1|t}^{d,\theta}(X_i(\mathbf{t}_i)^d|X_i(\mathbf{t}_i))\Big\}.$$

Let $\{(\mathbf{t}_i', X_i'(\mathbf{t}_i'), X_i'(1))\}_{i=1}^{n_{\text{test}}}$ be a test dataset independent of $\mathbb{D}_n$. To evaluate the estimation error of the estimated transition rate $\hat{u}$, we use the following empirical prediction risk

$$-\frac{1}{n_{\text{test}}}\sum_{i=1}^{n_{\text{test}}}\sum_{d=1}^{\mathcal{D}}\frac{1}{1-\mathbf{t}_i'}\Big\{\Big(1 - \delta_{X_i'(1)^d}(X_i'(\mathbf{t}_i')^d)\Big)\log p_{1|t}^{d,\theta}(X_i'(1)^d|X_i(\mathbf{t}_i')) - \delta_{X_i(1)^d}(X_i'(\mathbf{t}_i')^d) + p_{1|t}^{d,\theta}(X_i'(\mathbf{t}_i')^d|X_i'(\mathbf{t}_i'))\Big\}.$$

We set $n_{\text{test}} = 100,000$ in our simulation.

**Models.** All our logit models use ReLU networks with 4 hidden layers with 256 dimensions. The optimizer is Adam with learning rate 1e-3. We train on the $\mathcal{D}$-dimensional dataset for $2000\mathcal{D}/3$ epochs with batch size 512.

### I.2. Overall Performance Evaluation

**Tau-leaping algorithm.** We present the tau-leaping algorithm (Algorithm 2) described in (Campbell et al., 2022).

---

**Algorithm 2** Sampling via Tau-leaping (Algorithm 1 in Campbell et al., 2022)

---

**Require:** A learned transition rate $\hat{u}$, an early stopping parameter $\tau > 0$, time partition $0 = t_0 < t_1 < \cdots < t_N = 1 - \tau$.
1: Draw $Y_0 \sim \mathcal{U}(\mathcal{S}^{\mathcal{D}})$.
2: **for** $k = 0$ to $N - 1$ **do**
3:     **for** $d = 1$ to $\mathcal{D}$ **do**
4:         **for** $s \in \mathcal{S} \backslash Y_k^d$ **do**
5:             Draw $P_{ds} \sim \text{Poisson}((t_{k+1} - t_k)\hat{u}_{t_k}^d(s, Y_k))$.
6:         **end for**
7:     **end for**
8:     **for** $d = 1$ to $\mathcal{D}$ **do**
9:         **if** $\sum_{s \in \mathcal{S} \backslash Y_k^d} P_{ds} > 1$ **then**
10:             $Y_{k+1}^d = Y_k^d$
11:         **else**
12:             $Y_{k+1}^d = Y_k^d + \sum_{s \in \mathcal{S} \backslash Y_k^d} P_{ds} \times (s - Y_k^d)$
13:         **end if**
14:     **end for**
15: **end for**
16: **return** $Y_N \sim \hat{p}_{1-\tau}$

---

**Implementation Details.** We train our models with sample size 100,000. To assess the performance of uniformization and tau-leaping sampling algorithms in practice, we calculate the total variation of the empirical joint distribution of the first 3 dimensions ($8^3 = 512$ states in total) between the samples from the true data distribution and the generated samples. We choose $N = 100$ and the time partition $t_i = (1 - \tau) \times i/N$ for both algorithms, and $\lambda_{k+1} = \mathcal{D}/(1 - t_{k+1})$ for uniformization algorithm. For evaluation, we generate 500,000 samples from the true data distribution and 100,000 samples from discrete flow-based models using different sampling algorithms. We also record the runtime of each algorithm for sampling 100,000 samples.

**Overall Performance and Comparison between Uniformization and Tau-leaping.** The simulation results are presented in Table 2. From the simulation results, we can obtain the following conclusions.

- As the early stopping parameter $\tau$ increases, the total variation decreases first and then increases. The minimum is achieved between $\tau = 0.01$ and $\tau = 0.07$.

- As $\tau$ decreases, the runtime of uniformization increases and that of tau-leaping is almost fixed. This is because in each time interval $[t_k, t_{k+1}]$, the number of function calls depends on $t_{k+1}$ for uniformization algorithm and is fixed for tau-leaping algorithm.

- The uniformization sampling algorithm performs well for moderately small $\tau$, and is sometimes worse than tau-leaping algorithm especially for extremely small $\tau$. One possible explanation is that the tau-leaping algorithm might reduce the large estimation error caused by extremely small $\tau$.

*Table 2.* Total variation (on the joint distribution of the first 3 dimensions) and runtime with uniformization and tau-leaping algorithms.

| $\mathcal{D}$ | $\tau$ | Total Variation | | Runtime (s) | |
|---|---|---|---|---|---|
| | | Uniformization | Tau-leaping | Uniformization | Tau-leaping |
| | 0.01 | **0.0670** | 0.0679 | 31.3777 | 10.6464 |
| | 0.03 | **0.0551** | 0.0561 | 30.3401 | 10.6581 |
| 3 | 0.05 | 0.0547 | **0.0534** | 29.1027 | 10.9488 |
| | 0.07 | **0.0590** | 0.0608 | 28.6259 | 10.6501 |
| | 0.1 | **0.0613** | 0.0684 | 27.7543 | 10.6748 |
| | 0.01 | **0.0588** | 0.0612 | 44.1496 | 13.3098 |
| | 0.03 | **0.0488** | 0.0516 | 40.6444 | 13.6291 |
| 6 | 0.05 | **0.0509** | 0.0522 | 40.3871 | 13.8540 |
| | 0.07 | **0.0594** | 0.0639 | 37.7639 | 13.4209 |
| | 0.1 | **0.0647** | 0.0683 | 37.7448 | 13.3129 |
| | 0.01 | 0.0643 | **0.0628** | 56.9279 | 16.0244 |
| | 0.03 | **0.0653** | 0.0666 | 53.7150 | 16.4715 |
| 9 | 0.05 | **0.0539** | 0.0581 | 56.2539 | 15.9569 |
| | 0.07 | **0.0581** | 0.0612 | 50.1394 | 17.3870 |
| | 0.1 | **0.0689** | 0.0719 | 50.7919 | 16.6256 |
| | 0.01 | 0.1061 | **0.0980** | 73.9635 | 18.8918 |
| | 0.03 | **0.0584** | 0.0607 | 64.0545 | 19.3838 |
| 12 | 0.05 | **0.0610** | 0.0640 | 64.8737 | 18.3714 |
| | 0.07 | **0.0647** | 0.0672 | 62.9912 | 18.3160 |
| | 0.1 | **0.0680** | 0.0723 | 57.3696 | 19.2553 |
| | 0.01 | **0.0816** | **0.0816** | 82.3939 | 21.9227 |
| | 0.03 | 0.0627 | **0.0621** | 73.7929 | 21.7484 |
| 15 | 0.05 | **0.0718** | 0.0729 | 70.4029 | 20.3761 |
| | 0.07 | **0.0757** | 0.0775 | 68.3231 | 20.5265 |
| | 0.1 | **0.0784** | 0.0803 | 65.7229 | 21.0171 |

