# OpenReview forum: "Error Analysis of Discrete Flow with Generator Matching"
_ICML.cc/2026/Conference — ICML 2026 regular_

### Official Review · Reviewer_UsJX · 2026-03-04

**Soundness:** 3
**Presentation:** 3
**Significance:** 3
**Originality:** 3
**Overall Recommendation:** 5
**Confidence:** 4

**Summary:**

This paper studies theoretical properties of discrete flow models. The authors leverage a Girsanov-type change-of-measure result for CTMC path measures to upper bound the KL divergence between marginals of two CTMCs. With this tool, the paper decomposes the distribution estimation error into (i) stochastic error, (ii) approximation error, and (iii) early stopping error. The main results provide non-asymptotic total variation (TV) bounds without imposing a lower-bound condition on oracle rates, and a faster TV rate under an additional uniform lower bound assumption on the oracle rates. Simulation experiments validate the theoretical results.

**Compliance With Llm Reviewing Policy:**

Affirmed.

**Final Justification:**

All of my concerns have been effectively addressed. The theoretical contribution, particularly the convergence order result for discrete flow models, is valuable and provides a solid foundation for future work in this area. I am willing to raise my score to 5 and recommend acceptance.

**Key Questions For Authors:**

1. When applying Theorem 8 (change of measure) to derive Theorem 1 and later bounds, how do you ensure the condition u^X=0 ⇒ u^Y=0 holds in the learning setting?
2. Could you explain how the theoretical results presented in this paper guide real-world applications?

**Limitations:**

None.

**Strengths And Weaknesses:**

Strengths:
This paper presents error analysis of discrete flow models, which is a timely contribution given the growing use of generative models. The error decomposition (stochastic + approximation + early stopping) provides a clean structure for analyzing the distribution error. The paper is overall well-written.

Weaknesses:
The experiments validate qualitative trends on synthetic simulations, but they do not demonstrate the implications for realistic discrete generative modeling tasks.

---

> ### Author Rebuttal · Authors · 2026-03-31
>
> We thank the reviewer for the valuable comments and suggestions. We appreciate the time you spent on the paper. We address your concerns below.
>
> **Q1**: *The experiments validate qualitative trends on synthetic simulations, but they do not demonstrate the implications for realistic discrete generative modeling tasks.*
>
> **A1**: Thanks for this comment. Our current analysis of the statistical convergence rate cannot be directly applied to large-scale complex tasks, since the components in the error bound are not only related to the estimation error but also to the optimization error, which is not studied in this work; we leave this important direction for future work.
>
> **Q2**: *When applying Theorem 8 (change of measure) to derive Theorem 1 and later bounds, how do you ensure the condition $u^X=0 \to u^Y=0$ holds in the learning setting?*
>
> **A2**: This is a great question. We use the clipping parameter $m_n$ in our function class for truncation (equation 7), which is positive and can go to zero as the sample size increases. Consequently, we have $\hat{u}_t(z,x) = 0 \to u^0_t(z,x) = 0$.
>
> **Q3**: *Could you explain how the theoretical results presented in this paper guide real-world applications?*
>
> **A3**: Thanks for your question. We discuss this limitation in A3 part of the reply to Reviewer RQvt.

---

> > ### Author Rebuttal · Reviewer_UsJX · 2026-04-01
> >
> > Thank you for your reply. All my concerns have been well resolved.

---

### Official Review · Reviewer_GyqX · 2026-03-12

**Soundness:** 3
**Presentation:** 3
**Significance:** 3
**Originality:** 3
**Overall Recommendation:** 5
**Confidence:** 4

**Summary:**

This paper develops a unified framework for analyzing the convergence properties of discrete flow models and establishes non-asymptotic error bounds in total variation distance. The analysis decomposes the total error into stochastic error, approximation error, and early-stopping error, leading to a convergence rate of $(n \tau^2)^{-\frac{\beta}{2(2\beta + \mathcal{D} + 1)}}$. Moreover, under an additional lower-bound assumption on the flow, the rate can be improved to $(n \tau^2)^{-\frac{\beta}{2\beta + \mathcal{D} + 1}}$.

**Compliance With Llm Reviewing Policy:**

Affirmed.

**Final Justification:**

This is a good paper, and I will maintain my current score.

**Key Questions For Authors:**

1. The convergence rate of discrete flow models is much slower than that of continuous models (which is $O(n^{-1})$). Does this imply that discrete flow models are much inferior to continuous ones?

**Limitations:**

yes

**Strengths And Weaknesses:**

Strengths
1. This paper presents a unified framework for deriving error bounds for discrete flow models. The mathematical derivations are solid and rigorous, and the approximation error is explicitly analyzed, which has been largely ignored in prior work.
2. The paper is well written and easy to follow.
3. Numerical experiments are provided to support and validate the theoretical error bounds.

Weaknesses

1. It would be helpful to include a comparison table summarizing the assumptions and error rates of this work versus previous methods, so that the improvements can be more clearly highlighted.
2. A comprehensive notation table would improve readability.
3. In practical applications, the proposed sampling method requires $M \times N$ function evaluations, making it much slower than $\tau$-leaping. It would therefore be valuable to incorporate more efficient sampling methods (e.g., $\tau$-leaping) into the framework and derive corresponding error bounds.
4. Because the uniformization method requires an upper-bounded Poisson rate, the paper assumes an upper-bounded flow. However, the flow in typical models is not bounded as $t \to 1$. Although the rate diverges, the sampling process is not actually singular: at least one jump occurs, which drives $x_t$ into the support of $p_{data}$. This suggests that the framework could be refined in this regime to obtain more accurate results, rather than relying solely on early stopping.
5. In practice, the training dataset is finite and the model is trained to fit this empirical distribution. Thus $p_{data}$ represents the distribution of the collected data and is reachable, which makes the stochastic error less critical in many practical scenarios.

Minor:

1. $\mathcal{X}$ in Definition 3 is not defined.
2. In line 1107, the first term may be missing a factor of 2.

Overall, this is a good paper, but it would be strengthened by incorporating more efficient samplers into the framework and addressing the near-singular behavior as $t \to 1$ more directly, rather than just using early stopping.

---

> ### Author Rebuttal · Authors · 2026-03-31
>
> We thank the reviewer for the valuable comments and suggestions. We appreciate the time you spent on the paper. We address your concerns below. (We answer the reviewer's questions in the same top-to-bottom order as they appear in the reviews.)
>
> Answer for Weakness:
>
> **A1**: We thank the reviewer for this suggestion. We will add a comparison table in the revision, regarding convergence rate, state space, training objective, and whether taking early stopping into account.
>
> **A2**: Thanks for your suggestion. We will add a notation table in the revision.
>
> **A3**:  We discuss this limitation in A4 part of the reply to Reviewer RQvt.
>
> **A4**: Thanks for your question. Firstly, we want to mention that the early stopping parameter is related to the stochastic error, and the stochastic error is not related to the specific sampler used in the sampling stage. Without early stopping, the expected loss would not exist typically, since $\mathbb{E}  |\mathcal{L}(u)|=\mathbb{E} \{\frac{\dot{\kappa} _t}{1-\kappa _t}\Big|\sum _{s\neq X(\mathbf{t})}(\log u _t(s,X(\mathbf{t}))+u(s,X(\mathbf{t})))\Big|\}\ge C\mathbb{E} \{\frac{\dot{\kappa} _t}{1-\kappa _t}\}=\infty$ (for some constant $C$, where $\mathbf{t}\in\mathcal{U}([0,1])$ and we consider 1-$\mathcal{D}$ setting for simplicity). Secondly, the early stopping is not only a technique in discrete flow models, but also a standard component in the theoretical works on continuous flow matching (see, e.g. \cite{gao2024convergence,zhou2025an}). Thirdly, the early stopping is practically used in discrete flow-based models and masked diffusion (one can see Meta's flow matching repo \url{https://github.com/facebookresearch/flow_matching} and LLaDA training code: \url{https://github.com/ML-GSAI/SMDM}).
>
> **A5**: Thanks for your comment. When the goal is to fit the empirical distribution exactly (i.e., memorize training data or generate the samples from the empirical distribution), the stochastic error is less relevant (since you could generate infinite time variables) and approximation error might dominate. Our paper studies the standard generalization setting, where the target is the population distribution and the stochastic error is therefore essential.
>
> Answer for Minor:
>
> **A6**: Thank you for carefully reading. (1) $\mathcal{X}$ in Definition 3 should be $\mathcal{S}^\mathcal{D} \times [0,1]$ (the input domain). (2) The factor of 2 is not needed in the first term in line 1107. Since $\mathcal{L}(\hat{u})-\mathcal{L}(u^*)\leq 0$, the difference of the third term and the 4-th term would be less than zero in line 1105.
>
> Answer for Key Questions:
>
> **A7**: Thanks for your question. Firstly, the discrete state space and continuous state space are different, and it is infeasible and unreasonable to model discrete data on a continuous state space, so they cannot be directly compared. Secondly, the estimation error or distribution error of continuous flow matching actually cannot achieve the $O(n^{-1})$ rate, since one has to balance the stochastic error and approximation error (see [1] for a $O(n^{-1})$ stochastic error bound and [2] for a slow $O(n^{-1/2})$ rate with sample size). For the stochastic error, in our paper, we derive both slow rate and fast rate (which requires careful analysis and the boundedness condition due to the specific form of the Bregman divergence as we mentioned in Section 4.2). Thirdly, the estimation error bound (fast rate) for KL divergence (Equation 25) in our paper is actually aligned with the results ($O(n^{-2/(d+3)})$) derived in continuous flow matching (See Theorem 5.12 in [1]). Thus, our results do not suggest that discrete flow models are inferior to continuous ones.
>
> [1] Yuan Gao, Jian Huang, Yuling Jiao, and Shurong Zheng. Convergence of continuous normalizing flows for learning probability
> distributions. arXiv preprint arXiv:2404.00551, 2024
>
> [2] Zhengyu Zhou and Weiwei Liu. An error analysis of flow matching for deep generative modeling. In Forty-second International
> Conference on Machine Learning, 2025

---

> > ### Author Rebuttal · Reviewer_GyqX · 2026-04-04
> >
> > Thanks for your rebuttal. Most of my concerns have been satisfactorily addressed, so I’ve decided to keep my score.

---

### Official Review · Reviewer_RQvt · 2026-03-12

**Soundness:** 3
**Presentation:** 3
**Significance:** 3
**Originality:** 3
**Overall Recommendation:** 4
**Confidence:** 1

**Summary:**

This paper analyzes the statistical and approximation properties of discrete flow models trained via generator matching on finite-state continuous-time Markov chains (CTMCs). It establishes a unified theoretical framework based on a CTMC Girsanov-type theorem, deriving KL divergence bounds and non-asymptotic upper bounds for the model distribution error. The total error is decomposed into estimation, approximation, and early-stopping components, and faster nearly-optimal convergence rates are obtained under an oracle-rate assumption. Simulations verify the theoretical behavior with respect to sample size, dimension, and early-stopping parameters. This work offers formal theoretical characterizations for discrete flow models and their error structure compared with discrete diffusion models.

**Compliance With Llm Reviewing Policy:**

Affirmed.

**Key Questions For Authors:**

1. Could you extend your synthetic experiments to realistic discrete generative tasks to better validate the practical relevance of your theory?
2. Would you move key intuition and interpretability from the appendix into the main text to improve readability?
3. How can your theoretical rates and error decomposition guide practical choices such as model selection, early stopping, or sampler design?

**Limitations:**

1. Limited practical validation: The current empirical evidence mostly confirms qualitative trends on synthetic problems, not practical discrete generative modeling tasks.
2. Computational gap in sampling: Exact sampling via uniformization is theoretically appealing but may be too inefficient to represent realistic use cases.
3. Operational guidance remains weak: The theory explains what errors exist, but not yet how a practitioner should concretely use the results for tuning or model selection.

**Strengths And Weaknesses:**

Strengths
1. Discrete flow models have been gaining momentum, but their statistical guarantees are far less mature than those for continuous flow or some discrete diffusion variants. This paper directly tackles that gap.
2. The treatment of transition-rate estimation error is a real contribution. Many prior analyses effectively bury this issue inside stronger assumptions. Here, the paper makes the estimation problem explicit and threads it through the final distribution bound.
3. The three-way decomposition into stochastic, approximation, and early-stopping error is useful. This decomposition makes the final bounds more interpretable and potentially more actionable for both theorists and practitioners.
4. The theory is internally coherent. The sequence from CTMC preliminaries to KL control, then to risk decomposition, then to total-variation bounds is logically organized and technically aligned.


Weaknesses
1. The empirical section is too narrow for a top-tier ML venue. The experiments are synthetic sanity checks rather than evidence that the theory illuminates practical discrete-flow systems used in realistic generative modeling tasks.
2. The main paper is very dense and theorem-heavy. Important intuition, proof strategy, and practical interpretation are pushed into the appendix, which weakens accessibility.
3. The paper provides rates and decompositions, but gives little guidance on how these results should influence actual model selection, early stopping, or sampler design.

---

> ### Author Rebuttal · Authors · 2026-03-31
>
> We thank the reviewer for the valuable comments and suggestions. We appreciate the time you spent on the paper. We address your concerns below.
>
> **Q1 \& Limitation 1**: *Could you extend your synthetic experiments to realistic discrete generative tasks to better validate the practical relevance of your theory? Limited practical validation: The current empirical evidence mostly confirms qualitative trends on synthetic problems, not practical discrete generative modeling tasks.*
>
> **A1**: Thanks for your question. We agree that the current analysis cannot be directly applied to large-scale systems. Additionally, there is a gap between our theory and realistic systems, since our analysis mainly focuses on the ERM estimator and does not consider the optimization error, which is an important future direction.
>
> **Q2**: *Would you move key intuition and interpretability from the appendix into the main text to improve readability?*
>
> **A2**: Thanks for your suggestion. We will add valuable discussions during the rebuttal period and move the key discussion and interpretability from the appendix into the main text in the revision.
>
> **Q3 \& Limitation 3**: *How can your theoretical rates and error decomposition guide practical choices such as model selection, early stopping, or sampler design? Operational guidance remains weak: The theory explains what errors exist, but not yet how a practitioner should concretely use the results for tuning or model selection.*
>
> **A3**: Thanks for the question. As we mentioned above, we agree that the practical guidance for realistic scenarios and complex systems is limited for our theoretical work. Here are some choices suggested by our theory: (1) For architecture selection, the proof of Theorem 4 suggests choosing a network class with a width not diverging with the sample size (but should diverge with $\mathcal{D}$), and a depth diverging with $n$. (2) For early stopping parameter selection, the theory suggests choosing the parameter diverging with the sample size in a polynomial rate. However, these components would be related to optimization error in practice, so further investigation is needed.
>
> **Limitation 2**: *Computational gap in sampling: Exact sampling via uniformization is theoretically appealing but may be too inefficient to represent realistic use cases.*
>
> **A4**: Thanks for the question. We fully agree that analyzing efficient samplers is important. The main difference between the current analysis and previous works on discretization error for efficient samplers is that we take estimation error into account. Since we use uniformization to eliminate the discretization error, the function class we choose could be networks without time-Lipschitz regularity, which would be needed for bounding the discretization error for non-exact samplers. (One may wonder why we do not use the oracle transition rate to derive the discretization error bound, since the oracle transition rate is definitely time-Lipschitz. However, it is not feasible when taking estimation error into account, since if you use oracle transition rate to derive the discretization error bound (that is, to bound the error between CTMC with oracle rate and the CTMC with some step-wise time-homogeneous version of the oracle rate), then for deriving the overall bound, you have to bound the error between the CTMC with step-wise time-homogeneous rate in terms of the oracle rate and that in terms of the estimated rate, which cannot be analyzed under the empirical risk minimization framework). From our perspective, we think that there are two ways to analyze the efficient samplers. The first one is to select a function class with time-Lipschitz regularity due to the time discretization, and then bound the discretization error using similar techniques in the existing works on concrete score matching; the approximation error of the function class with time-Lipschitz regularity requires more careful analysis and we will pursue this in future work. The second method to derive the overall error bound with efficient samplers is to choose masked source and analyze the posterior $p _{1|t}$, since in this case the posterior is time-independent (see, e.g. [1]), and then the estimated transition rate $u _t(z,x)=\frac{\dot{\kappa} _t}{1-\kappa _t}(p _{1|t}(z|x)-\delta_x(z))$ is time-Lipschitz if we select a function class to approximate $p _{1|t}$ without time-embedding (in this case, the proof of deriving the estimation error is entirely similar). By the way, we also want to mention that the expected number of function evaluations of the uniformization algorithm would typically decrease as the number of time-grid points increases for fixed $\tau$ and $\mathcal{D}$, since we could choose a more accurate upper bound of the rate in each time interval as the time grid becomes finer.
>
> [1] Gat et al.. Discrete flow matching. NeuRIPS. 2024.

---

> > ### Author Rebuttal · Reviewer_RQvt · 2026-04-03
> >
> > The rebuttal usefully clarifies the intended scope of the theory, provides some additional discussion on architecture choice and early stopping, and explains the role of uniformization in the analysis.

---

### Official Review · Reviewer_K8xu · 2026-03-13

**Soundness:** 3
**Presentation:** 3
**Significance:** 3
**Originality:** 3
**Overall Recommendation:** 4
**Confidence:** 2

**Summary:**

The goal of the paper is to provide a complete error analysis for flow matching models defined over discrete state spaces. In this setting, the generative model is represented as a continuous-time Markov chain whose transition rates are learned from data.

To generate samples from the learned model, the authors use a uniformization technique for sampling the Markov chain. This approach avoids time-discretization errors by simulating the process exactly through a Poisson-driven jump process. However, this exactness comes at the cost of increased computational effort, since the number of Poisson jumps can become large when the upper bound on the transition rate is large.

Because the sampling step introduces no discretization error, the analysis focuses primarily on the transition rate estimation error and how it affects the generated Markov chain. The transition rates are learned using a generator matching loss, and the paper studies how the error in estimating these rates propagates to the final distribution produced by the model.

The main contribution is a non-asymptotic error analysis of the resulting model. The authors decompose the total error into several components, including stochastic error from empirical risk minimization, approximation error from the function class used to represent the rates, and early stopping error arising during training. They show how these errors can be balanced through the choice of the early stopping parameter. To the best of my understanding, this appears to be the first theoretical error analysis for discrete flow models.

**Compliance With Llm Reviewing Policy:**

Affirmed.

**Final Justification:**

This is a good paper, only my concerns about practical implications are not addressed.

**Key Questions For Authors:**

Please see above, specially regarding the implications of the error bounds on the algorithm design.

**Limitations:**

To some extent.

**Strengths And Weaknesses:**

__Soundness__: Overall, I found the paper to be technically sound to the extent that I checked the results. The theoretical development is careful and the level of mathematical rigor is high.

__Presentation and Significance__: I comment on presentation and significance together, because I believe that a clearer discussion of the results and modeling assumptions would significantly increase the perceived impact of the paper.

- First, the paper relies on several modeling assumptions that play a central role in the analysis, including uniformization for simulation, conditional independence assumptions, and sparsity of the transition rate. These assumptions are mathematically convenient, but their computational and practical implications are not discussed in sufficient detail. Expanding this discussion would help readers understand when the theory applies in practice and how these assumptions affect algorithm design.

- In particular, the conditional independence assumption raises several questions that could be explored further. For example, it would be helpful to clarify how this assumption affects the different components of the error bounds, especially the stochastic error term. It also raises the question of whether alternative constructions—perhaps computationally more demanding but statistically more efficient—could reduce the error. More broadly, it would be interesting if the analysis could provide guidance on what structure an optimal transition rate design should have, based on the derived bounds.

- Regarding significance, I do believe the paper addresses an important problem. Flow matching models and discrete diffusion models are increasingly relevant in modern generative modeling, and theoretical analysis of these models is valuable for both understanding their behavior and guiding algorithm design. However, the paper could do more to translate the theoretical conclusions into practical insights.

- For instance, the implications of Theorem 2 are not clearly discussed. It would strengthen the paper to explain what this theorem implies for the design choices practitioners make when constructing a flow matching model, such as architecture selection, regularization, or rate parameterization.

- A similar point applies to Theorem 3, which concerns approximation error. One might expect the complexity of the target distribution to influence this analysis, yet this connection is not clearly discussed

- Another aspect that deserves more discussion is the computational implication of the uniformization-based simulation scheme. While uniformization eliminates discretization error, it may lead to increased computational cost due to a large number of Poisson jumps when the rate bound is large. The conclusion briefly mentions this trade-off, but since uniformization is a central modeling component of the analysis, it would be beneficial to discuss its practical implications and possible alternatives in greater detail.

- Finally, I was somewhat unsure about the purpose of the numerical experiments. It is not entirely clear whether the experiments are intended to demonstrate that the learning framework works in practice or to empirically validate the theoretical error analysis. If the latter is the goal, it would be helpful to provide the curve you predict from the error analysis along with the empirical curves.

---

> ### Author Rebuttal · Authors · 2026-03-31
>
> We thank the reviewer for the valuable comments and suggestions. We appreciate the time you spent on the paper. We address your concerns below. (We answer the reviewer's questions in the same top-to-bottom order as they appear in the reviews.)
>
> **A1**: We thank the reviewer for this question. We clarify each assumption:
>
> (1) Conditional independence: See A2.
>
> (2) Sparsity of the transition rate: This is actually a direct consequence of the conditional independence. Additionally, as we mentioned in our paper, it reduces the rate matrix from $|\mathcal{S}|^{\mathcal{D}}$ entries to $\mathcal{D}(|\mathcal{S}|-1)+1$ non-trivial entries per state. For our bounds, the sparsity is why the stochastic error scales linearly in $\mathcal{D}$ rather than in $|\mathcal{S}|^\mathcal{D}$ (if we ignore the covering entropy).
>
> (3) Uniformization for simulation: There are two reasons for using uniformization: Firstly, we use this technique to bound the KL divergence between path measures of two CTMCs; secondly, uniformization yields zero discretization error, which simplifies our analysis. As an exact sampler, we agree that uniformization is computationally expensive (especially when the dimension is relatively high, as we mentioned in Section 6, and we view this as a limitation). We also discuss the limitation in A4 part of the reply to Reviewer RQvt.
>
> **A2**: This is the standard factorized conditional path $p_{t|1}(x|x_1)=\prod_d p_{t|1}^d(x^d|x_1^d)$ adopted by most existing practical discrete flow-based model implementations [1][2][3]. We also want to emphasize that it does not imply marginal independence, since the learned marginal rate $u_t(z,x)$ has inter-dimensional dependencies through the posterior $p_{1|t}(x_1|x)$. Additionally, this structure is fully parallel to the continuous flow matching where $X(t)$ is a convex combination of $X(1)$ and an isotropic Gaussian variable.
>
> **A3**: We appreciate this suggestion. We have to acknowledge that there is a gap between our theory and large-scale systems. Instead, we aim to provide a theoretical framework that captures the key error sources and reveals how these components potentially affect the convergence, and also provides large-sample convergence guarantees. Additionally, the error bound also reflects why the sparsity structure of the rate matrix is critical for convergence, since, if the rate matrix is dense, the stochastic error would scale exponentially in $\mathcal{D}$.
>
> **A4**: Thanks for the suggestion. Theorem 2 implies that the stochastic error of the ERM estimator scales with several components including covering entropy of the function class, dimension, early stopping and clipping parameter. For architecture selection, the proof of Theorem 4 suggests choosing a network class with a width not diverging with the sample size and a depth diverging with $n$. For early stopping and clipping parameter selection, combining the early stopping error, the theory suggests choosing the clipping parameter $m_n$ to have the order of $O(\tau^2)$ (if we have a fixed $\mathcal{D}|\mathcal{S}|$), since we choose $m_n$ to be our approximation error in Theorem 3. Finally, we acknowledge that the current work focuses only on convergence rates and therefore provides limited theoretical guidance on parameter tuning.
>
> **A5**: Thanks for the question. The target distribution has an indirect impact on our approximation error analysis since the target distribution will determine the property of the transition rate. As we mentioned in Section C1, when the target distribution has full support and the time schedule is linear, the rate will satisfy the boundedness condition, and the clipping parameter could be a constant instead of going to zero for truncation, which yields a smaller approximation error.
>
> **A6**: Thanks for the question. Tau-leaping is a more efficient sampler, which has a constant number of function evaluations as dimension increases if the time discretization scheme is fixed. When $t$ is close to 1, sampling with uniformization is computationally expensive.  A practical strategy is to use uniformization when $t$ is not extremely close to 1 and switch to tau-leaping when $t$ is close to the terminal time. We aim to extend our framework to efficient samplers in future work.
>
> **A7**: Thanks for the suggestion. The goal of our experiments is to validate the theoretical predictions of our error bound, and we will add the predicted curve in the revision.
>
>
> [1] Campbell et al. Generative flows on discrete state-spaces: Enabling multimodal flows with applications to protein co-design. ICML. 2024.
>
> [2] Gat et al.. Discrete flow matching. NeuRIPS. 2024.
>
> [3] Shaul et al. Flow matching with general discrete paths: a kinetic-optimal perspective. ICLR, 2025.

---

> > ### Author Rebuttal · Reviewer_K8xu · 2026-04-03
> >
> > Thank you for the response. This is a good paper, but my concerns regarding translating the theoretical bounds into practical implications are still there. I understand and respect that the authors choose not to do it in this paper .

---

### Decision · Program_Chairs · 2026-04-30

**Decision:**

Accept (regular)

**Comment:**

This paper develops non-asymptotic error analysis for discrete flow models trained with generator matching on discrete state spaces. By decomposing the estimation error into three interpretable components, the authors derive convergence rates both with and without boundedness conditions on the oracle transition rates.

The reviewers agree that the paper fills a theoretical gap and provides a technically sound analysis, which are recognized as meaningful contributions. The main recurring concern shared by the reviews is that the theoretical results offer limited practical guidance. The authors acknowledged these limitations and outlined potential paths forward in their rebuttal, which reviewers generally found satisfactory. Given the strength and timeliness of the theoretical contribution and the uniform support from the reviewers, I recommend acceptance.